# Bayes Adaptive Monte Carlo Tree Search for Offline Model-based Reinforcement Learning

**Jiayu Chen**
The University of Hong Kong
INFIFORCE Intelligent Technology
Hong Kong SAR
`jiayuc@hku.hk`

**Le Xu**
Tsinghua University
Beijing, China

**Wentse Chen & Jeff Schneider**
Carnegie Mellon University
Pittsburgh, PA 15213, USA

## Abstract

Offline reinforcement learning (RL) is a powerful approach for data-driven decision-making and control. Compared to model-free methods, offline model-based reinforcement learning (MBRL) explicitly learns world models from a static dataset and uses them as surrogate simulators, improving the data efficiency and enabling the learned policy to potentially generalize beyond the dataset support. However, there could be various MDPs that behave identically on the offline dataset and dealing with the uncertainty about the true MDP can be challenging. In this paper, we propose modeling offline MBRL as a Bayes Adaptive Markov Decision Process (BAMDP), which is a principled framework for addressing model uncertainty. We further propose a novel Bayes Adaptive Monte-Carlo planning algorithm capable of solving BAMDPs in continuous state and action spaces with stochastic transitions. This planning process is based on Monte Carlo Tree Search and can be integrated into offline MBRL as a policy improvement operator in policy iteration. Our "RL + Search" framework follows in the footsteps of superhuman AIs like AlphaZero, improving on current offline MBRL methods by incorporating more computation input. The proposed algorithm significantly outperforms state-of-the-art offline RL methods on twelve D4RL MuJoCo tasks and three challenging, stochastic tokamak control tasks.

## 1 Introduction

The success of RL typically relies on large amounts of interactions with the environment. However, in real-world scenarios, such interactions can be unsafe or costly. As an alternative, offline RL (Levine et al., 2020) leverages offline datasets of transitions, collected by a behavior policy, to train a policy. To avoid overestimation of the expected return for out-of-distribution states, which can mislead policy learning, model-free offline RL methods (Kumar et al., 2020; Wu et al., 2019) often constrain the learned policy to remain close to the behavior policy. However, acquiring a large volume of demonstrations from a high-quality behavior policy, can be expensive. This challenge has led to the development of offline model-based reinforcement learning (MBRL) approaches, such as Lu et al. (2022); Guo et al. (2022). These methods train dynamics models from offline data and optimize policies using imaginary rollouts simulated by the models. A key advantage of this approach is that the dynamics modeling process is decoupled from the behavior policy's objectives. This makes it possible, in principle, to learn effective policies even from sub-optimal or exploratory data, so long as the dataset provides adequate coverage of the relevant state-action space.

Multiple MDPs can exhibit identical behaviors on an offline dataset of states and actions, yet their underlying dynamics and reward functions may differ, particularly on out-of-sample regions. This implies that we are dealing with a distribution of possible world models underlying a dataset. A common strategy in offline MBRL is to learn an ensemble of world models and treat them equally. For instance, when determining the next state, a world model would be uniformly sampled from the ensemble to generate its prediction. However, **different ensemble members may perform better in different regions of the state-action space, making it necessary to adapt the belief over each ensemble based on the experience.** The Bayes Adaptive Markov Decision Process (BAMDP) (Duff, 2002) provides a principled framework for modeling such adaptations, and BAMCP (Guez

et al., 2013) is an efficient online planning method for solving BAMDPs. However, BAMCP has several limitations: (1) it relies on a ground-truth world model for planning; (2) it is restricted to discrete state and action spaces; and (3) its outcome is an action choice at a particular state, rather than a policy function. To address these challenges: (1) we learn an ensemble of world models from the offline dataset and construct a pessimistic BAMDP as an estimation of the true MDP; (2) we propose a novel planning algorithm to solve BAMDPs in continuous state and action spaces based on double progressive widening (Auger et al., 2013); and (3) we distill the planning outcomes into a policy function via RL, enabling real-time execution. **Notably, we provide a theoretical proof establishing the consistency of our planner in continuous, Bayes-adaptive MDP settings.**

Our main contributions include: (1) Firstly using BAMDPs to handle model uncertainties in offline MBRL; (2) Proposing theoretically-grounded Continuous BAMCP for planning in continuous, stochastic BAMDPs; (3) Developing the first algorithm to successfully integrate Bayesian RL, offline MBRL, and deep search for data-driven continuous control; (4) Demonstrating the state-of-the-art performance on twelve D4RL MuJoCo tasks and three target tracking tasks in tokamak control (for nuclear fusion).

## 2 BACKGROUND

An MDP (Puterman, 2014) can be described as a tuple $\mathcal{M} = \langle \mathcal{S}, \mathcal{A}, \mathcal{P}, \mathcal{R}, \gamma \rangle$. $\mathcal{S}$ and $\mathcal{A}$ are the state space and action space, respectively. $\mathcal{P} : \mathcal{S} \times \mathcal{A} \to \Delta_{\mathcal{S}}$ is the dynamics function and $\mathcal{R} : \mathcal{S} \times \mathcal{A} \to \Delta_{[0,1]}$ is the reward function, where $\Delta_{\mathcal{X}}$ denotes the set of probability distributions on $\mathcal{X}$. $\gamma \in [0, 1)$ is a discount factor. A Bayes Adaptive MDP (BAMDP) (Duff, 2002) can model scenarios where the precise MDP $\mathcal{M}_\theta = \langle \mathcal{S}, \mathcal{A}, \mathcal{P}_\theta, \mathcal{R}_\theta, \gamma \rangle$ is uncertain but is known to follow a prior distribution $b_0(\theta)$. During planning, a Bayes-optimal agent would update its belief over the MDP based on its experience. Formally, a BAMDP can be described as a tuple $\mathcal{M}^+ = \langle \mathcal{S}^+, \mathcal{A}, \mathcal{P}^+, \mathcal{R}^+, \gamma \rangle$. $\mathcal{S}^+$ denotes the space of information states $(s, b)$, which is a composition of the physical state and the current belief over the MDP. After each transition $(s, a, r, s')$, the belief is updated to the corresponding Bayesian posterior: $b'(\theta) \propto b(\theta)P((s, a, r, s')|\theta) = b(\theta)\mathcal{P}_\theta(s'|s, a)\mathcal{R}_\theta(r|s, a)$. Accordingly, $\mathcal{P}^+$ and $\mathcal{R}^+$ are defined as follows:

$$\mathcal{P}^+((s', b'')|(s, b), a) = \mathbb{1}(b'' = b') \int_\theta \mathcal{P}_\theta(s'|s, a)b(\theta)d\theta$$

$$\mathcal{R}^+((s, b), a) = \int_\theta \mathcal{R}_\theta(s, a)b(\theta)d\theta \tag{1}$$

The Q-function that satisfies the Bellman optimality equations (i.e., Eq. (2)) is the Bayes-optimal Q-function and $\pi^*(s, b) = \arg\max_a Q^*((s, b), a)$ is the Bayes-optimal policy. Actions derived from $\pi^*$ are executed in the true MDP and constitute the best course of actions for a Bayesian agent with a prior belief $b_0$ over the underlying MDP (Guez et al., 2014). Bayesian RL (Ghavamzadeh et al., 2015) seeks to compute the optimal policy, $\pi^*$, and provides a principled framework for handling uncertainty in the world model $\mathcal{M}_\theta$. However, the requirement for Bayesian posterior updates (i.e., Eq. (1)) often renders it computationally intractable, and we discuss several approximate methods in Section 3.

$$Q^*(x, a) = \mathcal{R}^+(x, a) + \gamma \int_{x'} V^*(x')\mathcal{P}^+(x'|x, a)dx'$$

$$V^*(x') = \max_a Q^*(x', a), \ \forall x = (s, b) \in \mathcal{S}^+, \ a \in \mathcal{A} \tag{2}$$

## 3 RELATED WORKS

**Offline model-based RL:** Offline RL (Chen et al., 2024) enables an agent to learn control policies from datasets of environment transitions pre-collected by a behavior policy $\mu$, i.e., $\mathcal{D}_\mu = \{[(s_t^i, a_t^i, r_t^i)_{t=1}^T]_{i=1}^N\}$. Offline Model-based RL (MBRL) methods explicitly learn world models $\mathcal{M}_\theta$ from $\mathcal{D}_\mu$ and adopt $\mathcal{M}_\theta$ as a surrogate simulator, enabling the learned policy to possibly generalize to states beyond $\mathcal{D}_\mu$. Both planning methods (Argenson & Dulac-Arnold, 2021; Zhan et al., 2022; Diehl et al., 2023) and RL methods (Yu et al., 2020; Kidambi et al., 2020; Lu et al., 2022) can use the learned $\mathcal{M}_\theta$ to obtain a policy. However, since $\mathcal{D}_\mu$ may not span the entire state-action space, $\mathcal{M}_\theta$ is unlikely to be globally accurate. Learning/Planning without any safeguards against such

model inaccuracy can yield poor results. Yu et al. (2020); Kidambi et al. (2020); Lu et al. (2022) propose learning an ensemble of world models, using ensemble-based uncertainty estimations to construct a pessimistic MDP (P-MDP), and learning a near-optimal policy atop it. Ideally, for any policy, the performance in the real environment is lower-bounded by the performance in the corresponding P-MDP (with high probability), thus avoiding being overly optimistic about an inaccurate model. **Notably, none of these offline MBRL methods have modeled the problem as a BAMDP, although Bayesian RL provides a principled framework for handling model uncertainty.**

**Bayes-adaptive planning:** Approximate methods for solving BAMDPs, such as Asmuth et al. (2009); Sorg et al. (2010); Castro & Precup (2010); Asmuth & Littman (2011); Wang et al. (2012); Guez et al. (2013); Slade et al. (2020) have been developed. As a representative work, BAMCP (Guez et al., 2013) adopts Monte-Carlo Tree Search (MCTS) (Browne et al., 2012) for Bayes-adaptive planning and is shown to converge in probability to a near Bayes-optimal policy at the root node of the search tree. **However, all these methods cannot be applied to large-scale MDPs with continuous state and action spaces.**

Also, these planning algorithms are not designed for offline MBRL. How to incorporate planning outcomes for policy improvement in RL and how to handle the inaccuracy of the learned world model during planning still require exploration.

## 4 METHODOLOGY

In this section, we present a novel offline MBRL algorithm based on Bayes Adaptive MCTS. In Section 4.2, we propose a Bayes Adaptive planning method (i.e., Continuous BAMCP) that can be applied to stochastic continuous control tasks. Then, in Section 4.3, we present a search-based policy iteration framework, where the search results (from Continuous BAMCP) are distilled into policy and value networks for policy improvement/evaluation at each iteration. In this way, we integrate offline MBRL with Bayes Adaptive MCTS. Both components require the use of an ensemble of world models for either practical implementation or uncertainty quantification, as detailed in Section 4.1.

### 4.1 THE KEY ROLE OF DEEP ENSEMBLES

Offline MBRL methods estimate world models $\mathcal{M}_\theta$ from a static dataset $\mathcal{D}_\mu$. However, there could be various MDPs that behave identically on the limited set of states and actions in $\mathcal{D}_\mu$, but their dynamics and reward functions may differ, especially on out-of-sample states and actions. Thus, we are actually dealing with a distribution of world models that follow a prior distribution $b_0(\theta) \triangleq P(\mathcal{M}_\theta | \mathcal{D}_\mu)$. As introduced in Section 2, Bayesian RL handles such model uncertainty by explicitly including the belief over the models in its state representation. The belief is updated with experience, providing a measure of how the models' uncertainty has changed since the beginning of the episode.

The idea of deep ensembles (Lakshminarayanan et al., 2017) is to train multiple deep neural networks as approximations of a function (for uncertainty quantification), each using a different weight initialization and optimized with a different mini-batch sequence. For offline MBRL, we can learn an ensemble of dynamics models $\{\mathcal{P}_\theta^1, \cdots, \mathcal{P}_\theta^K\}$ and reward models $\{\mathcal{R}_\theta^1, \cdots, \mathcal{R}_\theta^K\}$ from the dataset $\mathcal{D}_\mu$ by minimizing the following supervised learning loss: $(i = 1, \cdots, K)$

$$\mathcal{L}(\mathcal{P}_\theta^i) = -\mathbb{E}_{(s,a,s')\sim\mathcal{D}_\mu} \left[ \log \mathcal{P}_\theta^i(s'|s,a) \right]$$
$$\mathcal{L}(\mathcal{R}_\theta^i) = -\mathbb{E}_{(s,a,r)\sim\mathcal{D}_\mu} \left[ \log \mathcal{R}_\theta^i(r|s,a) \right] \tag{3}$$

$\{(\mathcal{P}_\theta^i, \mathcal{R}_\theta^i)_{i=1}^K\}$ **can be viewed as a set of independent and identically distributed (IID) samples from the prior** $P(\mathcal{M}_\theta | \mathcal{D}_\mu)$ **and constitute a finite approximation of the space of world models.** With such an ensemble, the belief over the world models can be converted to a mass function over a set of $K$ items, where the $i$-th element denotes the probability of being in the MDP $(\mathcal{P}_\theta^i, \mathcal{R}_\theta^i)$. In this case, a reasonable prior distribution is $b_0(\theta) = [1/K, \cdots, 1/K]$, since these models are IID prior samples. After receiving a transition $(s, a, r, s')$, the belief can be updated as follows:

$$b'(\theta)(i) \propto b(\theta)(i)\mathcal{P}_\theta^i(s'|s,a)\mathcal{R}_\theta^i(r|s,a), \; i = 1, \cdots, K \tag{4}$$

This is a practical implementation of the Bayesian posterior (originally defined in Eq. (1)) update based on deep ensembles, where $b(\theta)$, $b'(\theta)$, and $\mathcal{P}_\theta^i(s'|s,a)\mathcal{R}_\theta^i(r|s,a)$ denote the prior, posterior distributions, and likelihood, respectively.

Although the agent could adapt its belief and follow more reliable ensemble members in the Bayesian RL framework, there could be regions in the state-action space where none of the members generalize well, as they are all learned from a static offline dataset. A typical solution for such uncertainty is to construct a pessimistic MDP, which discourages the agent from exploiting regions with high uncertainty. **Another use of the deep ensemble is to construct a Pessimistic Bayes-Adaptive MDP (P-BAMDP).**

In particular, we apply a reward penalty to each transition $(s, a, r)$ and get a penalized reward $\tilde{r}$ as follows:

$$r - \lambda \cdot \text{std} \left[ r^i + \gamma \mathbb{E}_{s'^i \sim \mathcal{P}_\theta^i, a' \sim \pi(s'^i)} [Q_{\psi^-}(s'^i, a')] \right]_{i=1}^K \tag{5}$$

Here, $s'^i \sim \mathcal{P}_\theta^i(\cdot|s, a)$ and $r^i = \mathbb{E}[\mathcal{R}_\theta^i(\cdot|s, a)]$ denote the next state and expected reward from the $i$-th ensemble member. $\pi$ and $Q_{\psi^-}$ represent the policy and target Q-network. This penalty term is the standard deviation of the one-step lookahead Q-value targets, computed across all ensemble members. The underlying intuition is that a high variation in these predictions indicates a high degree of uncertainty at this region. As demonstrated by Sun et al. (2023), employing this penalty to define a pessimistic MDP yields an agent performance estimate that more closely matches real-world outcomes, compared to penalties based solely on ensemble disagreement in next-state prediction, as proposed in Yu et al. (2020); Kidambi et al. (2020).

## 4.2 Continuous BAMCP

BAMCP (Guez et al., 2013) has been successful in solving large-scale BAMDPs, **as detailed in Appendix A**, but it is limited to scenarios with discrete state and action spaces. In this subsection, we introduce a novel planning method to approximate the Bayes-optimal policy at a decision point $(s, h)$ ($h$ denotes the transition history that ends at $s$), which can be used to solve BAMDPs with continuous states/actions and stochastic transition kernels.

**Double Progressive Widening (DPW):** DPW (Couëtoux et al., 2011; Auger et al., 2013) is a technique to extend the use of MCTS to continuous state and action spaces. Instead of exploring all possible actions and next states, DPW maintains a finite list of options to search at each decision node, incrementally adding new options to the list based on the visitation counts of that decision node. To be specific, for a node $(s, h)$, a new action $a$ is sampled (with the current policy $\pi$) and added to its children set $C((s, h))$, if $\lfloor N((s, h))^\alpha \rfloor \geq |C((s, h))|$, where $\alpha \in (0, 1)$ is a hyperparameter that controls the growth rate and $N$ denotes the visitation counts of $(s, h)$. Otherwise, an action is selected from existing options in $C((s, h))$ according to the UCT (Kocsis & Szepesvári, 2006) rule. Similarly, to handle the infinitely many possible state transitions in stochastic environments, a new next state $s'$ is added to the children set $C((s, h), a)$ only if $\lfloor N((s, h), a)^\beta \rfloor \geq |C((s, h), a)|$ ($\beta \in (0, 1)$). Otherwise, the least visited child in $C((s, h), a)$ will be selected as the next state. **With DPW, the sets of possible actions or next states to explore are finite, allowing deep tree search as in discrete scenarios.**

**Integration of DPW and BAMCP:** Directly combining DPW and BAMCP (i.e., Algorithm 3) cannot solve BAMDPs with continuous state and action spaces. As introduced in Appendix A, BAMCP relies on root sampling, which samples dynamics functions only at the root node and follows a specific dynamics function throughout a simulation rollout. However, the rationale of root sampling (i.e., Lemma A.1) does not hold when applying DPW[1]. As an alternative design, Polynomial Upper Confidence Tree (PUCT) (Auger et al., 2013), built upon DPW, is a provably consistent planning method for solving MDPs with infinite-scale state and action spaces and highly stochastic transition dynamics. Thus, we propose to recast P-BAMDPs as MDPs and solve them using PUCT, as a novel Bayes planning method for continuous state and action spaces.

**Proposed algorithm – Continuous BAMCP:** In Algorithm 1, each simulation follows a path from the root node to an unvisited node, utilizing progressive widening when sampling actions or next states, as detailed in the ACTIONPW and STATEPW procedures. Compared to PUCT, the main modifications include: (1) replacing $\langle \mathcal{S}, \mathcal{P}, \mathcal{R} \rangle$ in MDPs with their extended definitions in BAMDPs,

---

[1]The last equality of Eq. (6) does not hold, since $\tilde{b}(\theta|has') \propto \tilde{b}(\theta|ha)\widetilde{\mathcal{P}}_\theta(s'|s, a) \neq \tilde{b}(\theta|ha)\mathcal{P}_\theta(s'|s, a)$. $\widetilde{\mathcal{P}}_\theta(s'|s, a)$ represents the distribution of next states when applying DPW, which differs from the true distribution $\mathcal{P}_\theta(s'|s, a)$, as dictated by the DPW rule.

---

**Algorithm 1** Continuous BAMCP

**Input:** $\pi, V, E, d_{\max}, \gamma, \alpha, \beta, \mathcal{P}_\theta^{1:K}, \mathcal{R}_\theta^{1:K}$
**procedure** SEARCH$((s,h), b(\theta))$
    **for** $e = 1 \cdots E$ **do**
        SIMULATE$((s,h), b(\theta), d_{\max})$
    **end for**
    $\pi_{\text{ret}}(a|(s,h)) \quad \propto \quad N((s,h),a)), a \quad \in \quad C((s,h)$
    $v_{\text{ret}} = \sum_{a \in C((s,h))} \frac{N((s,h),a)}{N((s,h))} Q((s,h),a)$
    **return** $\pi_{\text{ret}}, v_{\text{ret}}$
**end procedure**
**procedure** SIMULATE$((s,h), b(\theta), d)$
    **if** $d == 0$ **then return** $V((s,h))$
    $a \leftarrow$ ACTIONPW$((s,h))$
    $r, s', b'(\theta) \leftarrow$ STATEPW$((s,h), b(\theta), a)$
    $N((s,h)) += 1, N((s,h),a) += 1$
    **if** $N((s,h)) > 1$ **then**
        $R \leftarrow$ SIMULATE$((s', hars'), b'(\theta), d-1)$
    **else**
        $R \leftarrow V((s', hars'))$
    **end if**
    Access $\tilde{r}$ or calculate $\tilde{r}$ using Eq. (5)
    $R \leftarrow \tilde{r} + \gamma R$, cache $\tilde{r}$
    $Q((s,h),a) += \frac{R - Q((s,h),a)}{N((s,h),a)}$
    **return** $R$
**end procedure**

**procedure** ACTIONPW$((s,h))$
    **if** first visit **then** $C((s,h)) \leftarrow \emptyset$
    **if** $\lfloor N((s,h))^\alpha \rfloor \geq |C((s,h))|$ **then**
        $a \sim \pi(\cdot|(s,h))$
        $C((s,h)) \leftarrow C((s,h)) \cup \{a\}$
        $N((s,h),a), Q((s,h),a) \leftarrow 0, 0$
    **else**
        $a \leftarrow \arg\max_{x \in C((s,h))} \tilde{Q}((s,h),x)$
    **end if**
    **return** $a$
**end procedure**
**procedure** STATEPW$((s,h), b(\theta), a)$
    **if** first visit **then** $C((s,h),a) \leftarrow \emptyset$
    **if** $\lfloor N((s,h),a)^\beta \rfloor \geq |C((s,h),a)|$ **then**
        $r \sim \sum_{i=1}^K b(\theta)(i)\mathcal{R}_\theta^i(\cdot|s,a)$
        $s' \sim \sum_{i=1}^K b(\theta)(i)\mathcal{P}_\theta^i(\cdot|s,a)$
        Update $b(\theta)$ to $b'(\theta)$ using Eq. (4)
        $C((s,h),a) \leftarrow C((s,h),a) \cup \{(r, s', b'(\theta))\}$
        $N((s', hars')) \leftarrow 0$
        **return** $r, s', b'(\theta)$
    **end if**
    **return** the least visited node in $C((s,h),a)$
**end procedure**

---

i.e., $\langle \mathcal{S}^+, \mathcal{P}^+, \mathcal{R}^+ \rangle$, and (2) applying reward penalties to account for model uncertainty. To be specific, in STATEPW, $r$ and $s'$ are sampled from the distribution predicted by all ensemble members, which is a practical implementation of sampling from $\mathcal{R}^+$ and $\mathcal{P}^+$ as outlined in Eq. (1). After receiving the transition $(s, a, r, s')$, the belief vector $b(\theta)$ is updated to $b'(\theta)$ following Eq. (4), finishing the transition in $\mathcal{S}^+$ from $(s, b(\theta))$ to $(s', b'(\theta))$. The transition history $h$ is then updated to $h' = hars'$. Secondly, in SIMULATE, a penalized reward $\tilde{r}$ (i.e., Eq. (5)) is used, which effectively mitigates overfitting to inaccurate ensemble members by incorporating pessimism, as discussed in Section 4.1.

The following theorem establishes that our planner is a reliable and consistent solver for P-BAMDPs defined in Section 4.1. The detailed proof is provided in Appendix B. This consistency guarantee is crucial, as it ensures that with sufficient computation, our planner's output converges to the true optimal value within the pessimistic model, providing a reliable value target for our overall learning framework.

**Theorem 4.1.** *Let $\hat{V}(z_0)$ be the value of the root node $z_0$ estimated by **Continuous BAMCP** after $E$ simulations. Under the regularity and value function conditions defined in Appendix B.1, the error of this estimate with respect to the P-BAMDP's optimal value $V_p^*(z_0)$ is bounded as follows, exponentially surely in $E$:*

$$\left| \hat{V}(z_0) - V_p^*(z_0) \right| \leq C_0 E^{-\gamma_0} + \gamma^{d_{max}} \epsilon_V,$$

*where $\gamma_0, C_0 > 0$ are constants, $\gamma$ is the discount factor, $d_{max}$ is the maximum search depth, and $\epsilon_V$ is the uniform bound on the value approximation error at the leaf nodes.*

Algorithm 1 is used to approximate the Bayes-optimal policy[2] at $(s,h)$, which is $\pi_{\text{ret}}(a|(s,h)) \propto N((s,h),a), a \in C(s,h)$ (Auger et al., 2013). However, we aim to solve the entire P-BAMDP

---

[2]The value consistency guaranteed by Theorem 4.1 implies that the returned policy $\pi_{\text{ret}}$ is near-optimal for the P-BAMDP. Since our planner derives $\pi_{\text{ret}}$ through a pessimistic planning process, the pessimistic planning theory (Jin et al., 2020; Sun et al., 2023) ensures that a near-optimal policy in the P-BAMDP has a bounded

---

**Algorithm 2** BA-MCTS

---

**Input:** $T, E_l, \mathcal{P}_\theta^{1:K}, \mathcal{R}_\theta^{1:K}$
Initialize the actor $\pi$ and critic $Q_\psi$, $\mathcal{D} \leftarrow \emptyset$
**procedure** LEARNER
    $e \leftarrow 0$
    **while** true **do**
        $\{(s, h, \pi_{\text{ret}}, \tilde{r}, s', h')_i\}_{i=1}^B \sim \mathcal{D}$
        Update $\pi$ and $Q_\psi$ using SAC
        $e \mathrel{+}= 1$
        Update $\pi$ and $V$ (defined on $Q_\psi$) in
ACTOR if
        $e \% E_l == 0$
    **end while**
**end procedure**

**procedure** ACTOR
    **while** true **do**
        Sample $s$ from $\mathcal{D}_\mu$, $h \leftarrow s, \tau \leftarrow [\,]$
        Obtain the prior $b(\theta)$ at $s$
        **for** $t = 1 \cdots H$ **do**
            $\pi_{\text{ret}}, v_{\text{ret}} \leftarrow \text{SEARCH}((s, h), b(\theta))$
            $a \sim \pi_{\text{ret}}(\cdot|(s, h))$
            Acquire $r, s', b'(\theta)$ as in STATEPW
            Calculate $\tilde{r}$ using Eq. (5)
            Append $\tau$ with $((s, h), \tilde{r}, \pi_{\text{ret}}, v_{\text{ret}})$
            $s, h, b(\theta) \leftarrow s', hars', b'(\theta)$
        **end for**
        $\mathcal{D} \leftarrow \mathcal{D} \cup \{\tau\}$
    **end while**
**end procedure**

---

offline, eliminating the need for anything beyond simple inference using the policy network during deployment. This necessitates a well-learned policy function at each decision node, but we cannot execute Algorithm 1 at every $(s, h)$ due to the scale of the state space. Therefore, we integrate the planning algorithm into a policy iteration framework as introduced in the next subsection. In this case, $\pi$ and $V$ in Algorithm 1 denote the policy and value functions from the previous learning iteration[3]; while $\pi_{\text{ret}}$ and $v_{\text{ret}}$ are the improved policy and value estimates for specific decision nodes. As additional details, multiple terms (labeled in blue) in Algorithm 1 have alternative designs across different literatures, which we elaborate on in Appendix C.

### 4.3 OVERALL FRAMEWORK: BA-MCTS

Now, we present how to integrate planning outcomes of continuous BAMCP into a policy iteration framework, to obtain a near Bayes-optimal policy, i.e., $\pi$, that can be directly referred to during execution. The pseudo code is shown as Algorithm 2. For efficiency, a learner and a number of actors execute in parallel, reading from and sending data to the replay buffer $\mathcal{D}$ respectively. The actors update their copies of policy and value functions every $E_l$ learner steps.

Each actor interacts with the learned world models to sample trajectory segments $\tau$. The starting states of these segments are sampled from the provided dataset $\mathcal{D}_\mu$. Notably: (1) the segment length $H$ is kept short to minimize error accumulation when interacting with the learned world models; and (2) the prior $b(\theta)$ at a starting state $s$ is obtained by performing the Bayesian posterior update (i.e., Eq. (4)) on the offline trajectory in $\mathcal{D}_\mu$ containing $s$. These belief updates are reliable, as the offline trajectories are from the real environment. At each decision node $(s, h)$ of the segment, a SEARCH procedure (defined in Algorithm 1) is executed. The search result $\pi_{\text{ret}}$ then indicates the action choice, i.e., $a \sim \pi_{\text{ret}}(\cdot|(s, h))$. Given $a$, the transition to the next state follows a BAMDP, where $r \sim \mathcal{R}^+(\cdot|(s, h), a)$, $s' \sim \mathcal{P}^+(\cdot|(s, h), a)$, and the belief $b(\theta)$ is adapted with the new transition.

With the collected segments $\tau$, we update the policy $\pi$ and value function $V$. Our primary algorithm, BA-MCTS, employs an off-policy RL method: SAC (Haarnoja et al., 2018). The value function (critic) is updated using standard temporal difference learning based on the sampled transitions, while the policy (actor) is updated using the policy gradient derived from the critic. Additionally, we investigate an alternative approach, BA-MCTS-SL, where the policy is updated via supervised learning to mimic the search result $\pi_{\text{ret}}$ by minimizing a cross-entropy loss: $\mathcal{L}(\pi, \{(s, h, \pi_{\text{ret}})_i\}_{i=1}^B) = -\sum_{((s,h), \pi_{\text{ret}})} \pi_{\text{ret}}^T \log \pi(\cdot|(s, h))/B$. We compare these two approaches

---

sub-optimality in the non-pessimistic BAMDP. Finally, according to the foundational theory of BAMDPs Guez et al. (2013), a near-optimal policy for the BAMDP is, by definition, a near-Bayes-optimal policy for the real environment.

    [3] $\pi$ and $V$ are functions of $(s, h)$ because the states in BAMDPs include both $s$ and the corresponding belief $b(\theta)$. $b(\theta)$ is recursively updated using the history $h$ and is a function of $h$.

Table 1: Comparison of the prediction errors for next states and rewards in imaginary rollouts, with and without Bayesian belief adaptation. Ground truth for the imaginary rollouts is obtained by replaying the action sequences in the real simulators.

| Data Type | Environment | Prediction Error on Next State | | Prediction Error on Reward | | Overall Prediction Error | |
|---|---|---|---|---|---|---|---|
| | | Adaptive | Uniform | Adaptive | Uniform | Adaptive | Uniform |
| random | HalfCheetah | **0.339** (.021) | 0.342 (.017) | **0.016** (.001) | **0.016** (.001) | **0.177** (.011) | 0.179 (.009) |
| random | Hopper | 0.232 (.016) | **0.189** (.008) | **0.012** (.001) | 0.022 (.008) | 0.122 (.008) | **0.106** (.008) |
| random | Walker2d | **62.10** (11.0) | 112.0 (19.0) | **1.364** (.215) | 4.908 (.708) | **31.73** (5.61) | 58.44 (9.83) |
| medium | HalfCheetah | 1.387 (.040) | **1.355** (.038) | **0.057** (.001) | 0.061 (.001) | 0.722 (.021) | **0.708** (.019) |
| medium | Hopper | **0.429** (.033) | 0.570 (.035) | **19.03** (8.58) | 68.24 (11.4) | **9.727** (4.30) | 34.40 (5.69) |
| medium | Walker2d | **34.01** (.556) | 34.26 (.142) | **24.38** (4.31) | 113.1 (3.47) | **29.19** (1.89) | 73.69 (1.78) |
| med-replay | HalfCheetah | **0.677** (.027) | 0.707 (.036) | 0.115 (.005) | **0.114** (.006) | **0.396** (.016) | 0.410 (.021) |
| med-replay | Hopper | **0.170** (.022) | 0.212 (.054) | **1.177** (.420) | 3.320 (1.14) | **0.674** (.221) | 1.766 (.558) |
| med-replay | Walker2d | 66.83 (7.39) | **44.61** (1.34) | **27.21** (3.24) | 60.52 (3.17) | **47.02** (5.17) | 52.57 (2.25) |
| med-expert | HalfCheetah | **1.423** (.054) | 1.467 (.046) | **0.071** (.002) | 0.074 (.002) | **0.747** (.028) | 0.771 (.024) |
| med-expert | Hopper | **3.665** (2.56) | 18.27 (2.09) | **71.46** (79.9) | 384.7 (37.6) | **37.56** (41.2) | 201.5 (19.4) |
| med-expert | Walker2d | 55.69 (3.71) | **51.89** (.481) | **121.9** (12.6) | 186.3 (3.86) | **88.77** (8.18) | 119.1 (2.08) |

in our evaluation. In both methods, we seek to distill the superior, non-parametric search outcomes from Continuous BAMCP into the parameterized policy $\pi$.

## 5 EVALUATION

Section 5.1 discusses the benefit of Bayesian model adaptation. Section 5.2 presents comprehensive benchmarking results on D4RL MuJoCo tasks. Section 5.3 details our ablation studies. Finally, Section 5.4 validates our algorithm's applicability on a challenging Tokamak control task.

### 5.1 THE BENEFIT OF BELIEF ADAPTATION

The core idea of belief adaptation (using the BAMDP framework detailed in Section 4.1) is that by dynamically adjusting the belief over each ensemble member based on observed transitions, the agent can rely more on models that are accurate for the current trajectory, leading to better planning. Our empirical results show that belief adaptation not only improves model prediction accuracy, as shown in Table 1, but also translates to better final policy performance on offline RL tasks, as shown in Tables 2 and 4. Specifically, Table 1 highlights that the adaptive ensemble achieves more accurate predictions for next states and rewards in 10 out of 12 benchmarking tasks, compared to treating each ensemble member uniformly as in previous offline MBRL methods. This suggests that Bayesian belief adaptation leads to a more accurate world model for planning, which is a key factor underlying observed performance improvements. **A more detailed study of this aspect, including a visualization of the belief adaptation process within rollouts, is provided in Appendix F.1.**

### 5.2 BENCHMARKING RESULTS ON D4RL MUJOCO

We evaluate our algorithm on a widely-used offline RL benchmark: D4RL MuJoCo (Fu et al., 2020), comparing its performance with state-of-the-art (SOTA) model-based/model-free methods, and search-based approaches.

**Model-Based Methods.** In Table 2, BA-MCTS achieves the highest average score across all twelve D4RL MuJoCo tasks when compared against a range of SOTA model-based baselines. This result confirms that the combination of a principled Bayesian treatment of uncertainty and a powerful search-based planner leads to superior performance.

**Model-Free Methods. We also compare our algorithms with a series of model-free offline policy learning methods, with detailed results presented in Appendix F.2.** Our algorithms show significantly better performance, demonstrating the necessity of model-based learning in these environments, particularly when data quality is low. Notably, APE-V (see Table 2) is a model-free method built on BAMDPs, yet it shows inferior performance compared to ours in 10 out of 12 tasks. As a summary, in Table 3[4], we show comparisons against 19 strong model-based and model-free baselines, where BA-MCTS achieves the highest average score. (**See Appendix F.3 for details of the selected baselines.**) To quantify the significance of this improvement, we compute Cohen's $d$

---

[4]MOReL, IQL, and the model-free offline RL baselines in Appendix F.2 did not report standard deviations. Additionally, IQL and DT were not evaluated on D4RL tasks with random data.

Table 2: Comparisons of our algorithm against SOTA offline RL methods on the D4RL benchmark. Each value represents the normalized score, as proposed in Fu et al. (2020), of the policy trained by the algorithm. For our algorithm, we report the average score of the final ten learning epochs and its standard deviation across three random seeds. To ensure a fair comparison, the results for MOBILE and CBOP are from our own runs using their publicly codebases. The results for all other baselines are taken from their respective original papers. Please check Appendix F for the references.

| Data Type | Environment | BA-MCTS (ours) | MOBILE | CBOP | RAMBO | APE-V | MAPLE | Optimized |
|---|---|---|---|---|---|---|---|---|
| random | HalfCheetah | 41.45 (1.40) | 39.27 (1.38) | 32.95 (0.14) | 40.0 | 29.9 | **41.5** | 31.7 |
| random | Hopper | **31.42** (0.10) | 7.965 (0.43) | 23.79 (11.2) | 21.6 | 31.3 | 10.7 | 12.1 |
| random | Walker2d | 21.54 (0.06) | 21.47 (0.03) | 1.391 (0.06) | 11.5 | 15.5 | **22.1** | 21.7 |
| medium | HalfCheetah | 77.31 (1.40) | 76.42 (1.00) | 69.98 (0.88) | **77.6** | 69.1 | 48.5 | 45.7 |
| medium | Hopper | **103.9** (0.33) | 102.5 (1.77) | 98.68 (2.30) | 92.8 | - | 44.1 | 69.3 |
| medium | Walker2d | 88.23 (0.78) | 87.12 (2.79) | **93.92** (1.92) | 86.9 | 90.3 | 81.3 | 79.7 |
| med-replay | HalfCheetah | **71.24** (2.65) | 64.67 (8.14) | 63.36 (0.47) | 68.9 | 64.6 | 69.5 | 58.0 |
| med-replay | Hopper | **106.4** (0.53) | 105.7 (1.56) | 101.3 (1.74) | 96.6 | 98.5 | 85.0 | 90.8 |
| med-replay | Walker2d | **91.42** (1.54) | 85.12 (10.1) | 74.56 (1.29) | 85.0 | 82.9 | 75.4 | 65.8 |
| med-expert | HalfCheetah | 102.3 (2.27) | 100.9 (4.00) | 97.91 (5.26) | 93.7 | 101.4 | 55.4 | **104.2** |
| med-expert | Hopper | 111.8 (0.82) | **111.9** (0.34) | 111.2 (0.77) | 83.3 | 105.7 | 95.3 | 105.8 |
| med-expert | Walker2d | 116.0 (1.49) | 115.0 (0.30) | **117.6** (0.27) | 68.3 | 110.0 | 107.0 | 97.1 |
| Average Score | | **80.25** | 76.50 | 73.87 | 68.85 | 72.65 | 61.32 | 65.16 |

(Cohen, 1988) against all baselines for which standard deviations are available. The results show that the Cohen's $d$ value is greater than 1.8 in all cases. According to Cohen's Rule of Thumb, a value of $d > 1$ indicates a very large (statistically significant) difference. This comprehensive comparison demonstrates that our proposed method establishes a new SOTA in offline RL.

Table 3: Comprehensive performance comparison on D4RL MuJoCo. Average scores for our algorithm and baselines are reported. For baselines that report standard deviations, we compute Cohen's $d$ to quantify the statistical significance of the improvement achieved by our method. A value of $d > 1$ indicates a very large and statistically significant difference.

| Algorithm | BA-MCTS | Optimized | COMBO | MOReL | MOPO | CQL | BEAR |
|---|---|---|---|---|---|---|---|
| Avg. Score | **80.3 (1.1)** | 65.2 (5.4) | 66.8 (3.4) | 64.4 | 36.7 (11.0) | 54.9 | 39.8 |
| Avg. (No Random) | **96.5 (1.3)** | - | - | - | - | - | - |
| Cohen's $d$ | - | 3.5 | 4.6 | - | 4.7 | - | - |

| Algorithm | BRAC-v | SAC | BC | MOBILE | CBOP | RAMBO | APE-V |
|---|---|---|---|---|---|---|---|
| Avg. Score | 31.4 | 4.1 | 25.1 | 76.5 (2.7) | 73.9 (2.2) | 68.9 (2.8) | 72.7 (1.8) |
| Avg. (No Random) | - | - | - | - | - | - | - |
| Cohen's $d$ | - | - | - | 1.8 | 3.7 | 4.8 | 4.8 |

| Algorithm | MAPLE | TD3+BC | IQL | SAC-N | EDAC | DT | |
|---|---|---|---|---|---|---|---|
| Avg. Score | 61.3 (1.8) | 54.6 (2.4) | - | 76.3 (0.6) | 76.0 (2.3) | - | |
| Avg. (No Random) | - | - | 77.0 | - | - | 74.7 (1.2) | |
| Cohen's $d$ | 12.7 | 12.4 | - | 4.8 | 2.2 | 17.4 | |

**Search-Based Methods.** MuZero also applies deep search to MBRL. To evaluate its performance on D4RL MuJoCo, we use the open-source implementation and hyperparameter configurations of Sampled EfficientZero (Ye et al., 2021) provided by LightZero (Niu et al., 2023). Benchmarking results from LightZero indicate that Sampled EfficientZero achieves the best performance on (online) MuJoCo tasks compared to other MuZero variants. To adapt Sampled EfficientZero for offline learning, we employ the reanalyse technique proposed by Schrittwieser et al. (2021). **The evaluation results are presented in Figure 2 (Appendix F.4).** As shown, the results are significantly worse compared to the performance of offline RL methods listed in Table 2, despite Sampled EfficientZero's much higher computational cost. **In Appendix F.4, we provide a detailed comparison of the computational costs and algorithm designs between our method and Sampled EfficientZero.** World model learning is the foundation of MBRL and can be particularly challenging in continuous control and offline settings, where the state-action space is vast but training data is limited. Sampled EfficientZero integrates model learning and policy training into a single stage, which significantly increases the learning difficulty.

Table 4: Comparisons on the target tracking tasks. For each algorithm, we report the average return of the final ten policy learning epochs and its standard deviation across three different random seeds.

| Task | BA-MCTS (Ours) | BA-MCTS-SL (Ablation) | BA-MBRL (Ablation) | CQL | Optimized |
|---|---|---|---|---|---|
| Temperature | -23.83 (9.66) | **-21.16** (5.00) | -29.35 (4.72) | -59.62 (1.57) | -83.55 (10.56) |
| Rotation | -19.07 (5.85) | **-14.14** (1.88) | -31.33 (11.54) | -85.48 (2.72) | -71.54 (9.88) |
| $\beta_n$ | **-18.93** (1.75) | -37.03 (17.98) | -23.40 (10.77) | -36.37 (1.17) | -57.84 (10.27) |
| Average | **-20.61** | -24.11 | -28.03 | -60.49 | -70.98 |

## 5.3 ABLATION STUDY

For a principled ablation study, we reimplement BA-MCTS and its ablated variants based on the codebase of Optimized (Lu et al., 2022), which thoroughly explores design choices in offline MBRL, making minimal changes to the code and hyperparameter settings (see Appendix D). Specifically, we first add a belief adaptation module to Optimized to obtain **BA-MBRL**, and then incorporate the Continuous BAMCP planner (Section 4.2) to create **BA-MCTS**. Finally, we replace the policy distillation module of BA-MCTS from SAC to supervised learning (as detailed in Section 4.3), yielding **BA-MCTS-SL**. The detailed results are presented in Table 11, and Appendices F.5 - F.6. BA-MBRL significantly outperforms Optimized, demonstrating the necessity of Bayesian model adaptation. BA-MCTS further improves upon the performance of BA-MBRL, underscoring the effectiveness of employing deep search as a powerful policy improvement operator. Finally, BA-MCTS-SL achieves performance comparable to BA-MCTS. However, BA-MCTS-SL can be less stable on certain complex tasks: it struggles on the Walker2d environments, where a warm-up phase (using BA-MBRL) is needed to establish a good initial policy for supervised learning to be effective. Finally, we conduct an ablation study to assess the necessity of the reward penalty. By removing this penalty term (i.e., setting $\lambda = 0$ in Eq. (5)), we observe a consistent drop in average performance. This demonstrates that the pessimistic reward penalty is crucial for stopping the agent exploiting inaccuracies in the learned world models. **Results are in Table 12 in Appendix F.7.**

## 5.4 APPLICATION TO TOKAMAK CONTROL

We evaluate our algorithm on three target tracking tasks in tokamak control. The tokamak is one of the most promising confinement devices for achieving controllable nuclear fusion. We use a well-trained data-driven dynamics model provided by Char et al. (2024) as a "ground truth" simulator for the nuclear fusion process during evaluation, and generate a dataset (by this dynamics model) containing 725,270 transitions for offline RL. We select a reference shot (i.e., an episode of a fusion process) from DIII-D (a tokamak device located in San Diego), and use its trajectories of Ion Rotation, Electron Temperature, and $\beta_n$ as targets for three tracking tasks. The tracking tasks have a 28-dimensional state space and a 14-dimensional action space, both continuous. Moreover, these tasks are **highly stochastic**, as the underlying dynamics model is a probabilistic neural network and each state transition is a sample from this model. **For details on the simulator, and the design of the state/action spaces and reward functions, please refer to Appendix E**.

We obtained access to the DIII-D data for a limited time window, which restricted our ability to provide evaluation results across a wide range of algorithms. For a principled comparison with the baselines, we utilized the codebase from the ablation study, where our algorithm implementations are primarily based on Optimized, to ensure that performance improvements are attributable to algorithmic design rather than implementation differences. Additionally, we included evaluation of a widely-used model-free offline RL method: CQL. The average tracking errors over the final 10 training epochs are reported in Table 4, where BA-MCTS outperforms all baselines. These results demonstrate the potential of our algorithm for application in complex, data-driven control problems.

## 6 CONCLUSION

We propose framing offline MBRL as a BAMDP to better address uncertainties in the world models learned from offline datasets. We also introduce a novel planning method for solving BAMDPs in continuous state and action spaces using MCTS. This planning process is integrated into a policy iteration framework, enabling the derivation of a policy function for real-time execution. In our evaluation, we test several variants of our algorithms to separately highlight the effectiveness of Bayesian

RL and deep search. Additionally, we validate the superior performance of our approach on both standard offline RL benchmarks and challenging continuous control tasks in Nuclear Fusion.

## ACKNOWLEDGEMENT

This work was funded in part by Department of Energy Fusion Energy Sciences under grant DE-SC0024544.

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

# A BAMCP

---

**Algorithm 3** BAMCP

---

**Input:** $\pi, E, d_{\max}, \mathcal{R}, \gamma, \mathcal{A}, c$

**procedure** SEARCH$((s, h), b(\theta))$
    **for** $e = 1 \cdots E$ **do**
        $\theta \sim b(\cdot)$
        SIMULATE$((s, h), \theta, d_{\max})$
    **end for**
    **return** $\arg\max_a Q((s, h), a)$
**end procedure**

**procedure** ROLLOUT$((s, h), \theta, d)$
    **if** $d == 0$ **then**
        **return** $0$
    **end if**
    $a \sim \pi(\cdot|(s, h))$
    $s' \sim \mathcal{P}_\theta(\cdot|s, a)$
    $r \leftarrow \mathcal{R}(s, a)$
    $R \leftarrow r + \gamma$ROLLOUT$((s', has'), \theta, d-1)$
    **return** $R$
**end procedure**

**procedure** SIMULATE$((s, h), \theta, d)$
    **if** $d == 0$ **then**
        **return** $0$
    **end if**
    **if** $N((s, h)) == 0$ **then**
        **for** $a \in \mathcal{A}$ **do**
            $N((s, h), a), Q((s, h), a) \leftarrow 0, 0$
        **end for**
        $a \sim \pi(\cdot|(s, h))$
        $s' \sim \mathcal{P}_\theta(\cdot|s, a), r \leftarrow \mathcal{R}(s, a)$
        $R \leftarrow r + \gamma$ROLLOUT$((s', has'), \theta, d-1)$
        $N((s, h)), N((s, h), a) \leftarrow 1, 1$
        $Q((s, h), a) \leftarrow R$
        **return** $R$
    **end if**
    $a \leftarrow \arg\max_x Q((s, h), x) + c\sqrt{\frac{\log N((s, h))}{N((s, h), x)}}$
    $s' \sim \mathcal{P}_\theta(\cdot|s, a), r \leftarrow \mathcal{R}(s, a)$
    $R \leftarrow r + \gamma$SIMULATE$((s', has'), \theta, d-1)$
    $N((s, h)) \mathrel{+}= 1, N((s, h), a) \mathrel{+}= 1$
    $Q((s, h), a) \mathrel{+}= \frac{R - Q((s, h), a)}{N((s, h), a)}$
    **return** $R$
**end procedure**

---

Bayes Adaptive Monte Carlo Planning (BAMCP) (Guez et al., 2013) is a sample-based online planning method, aiming to find the action $a^*$ that approximately maximizes the expected return at a decision point $(s, h)$ under the BAMDP. Its detailed pseudo code is shown as Algorithm 3. BAMCP has demonstrated success in solving BAMDPs with large-scale **discrete** state and action spaces. Its key algorithmic ideas include: (1) applying MCTS with an efficient exploration strategy – UCT (Kocsis & Szepesvári, 2006) to the BAMDP in order to simulate the outcomes of different action choices; (2) utilizing root sampling to avoid frequent Bayesian posterior updates. Specifically, the UCT rule is used for selecting actions at non-leaf nodes, i.e., $a \leftarrow \arg\max_x Q((s, h), x) + c\sqrt{\frac{\log N((s, h))}{N((s, h), x)}}$, managing the tradeoff between exploration and exploitation. (As shown in Algorithm 3, $N$ is the visit count, while $Q$ denotes the accumulated reward.) Root sampling refers to sampling the dynamics model only at the root node (i.e., $\theta \sim b(\cdot)$) and not adapting the belief $b(\cdot)$ according to the Bayes rule and experience during the search process, of which the rationality is justified in the following lemma.

**Lemma A.1.** *For all suffix histories $h'$ of $h$, $b(\theta|h') = \tilde{b}(\theta|h')$. Here, $b(\theta|h')$ is the true posterior probability of $\theta$ at the decision point $h'$, while $\tilde{b}(\theta|h')$ is the probability of experiencing $\theta$ at $h'$ when using root sampling.*

*Proof.* This lemma can be proved by induction.

Base case: When $h' = h$, $b(\theta|h') = \tilde{b}(\theta|h') = b(\theta)$.

Step case:
$$
\begin{aligned}
b(\theta|has') &= P(has'|\theta)P(\theta)/P(has') \\
&= P(h|\theta)\mathcal{P}_\theta(s'|s, a)P(\theta)/P(has') \\
&= b(\theta|h)P(h)\mathcal{P}_\theta(s'|s, a)/P(has') \\
&= Zb(\theta|h)\mathcal{P}_\theta(s'|s, a) \\
&= Z\tilde{b}(\theta|h)\mathcal{P}_\theta(s'|s, a) \\
&= Z\tilde{b}(\theta|ha)\mathcal{P}_\theta(s'|s, a) = \tilde{b}(\theta|has')
\end{aligned}
\tag{6}
$$

Here, $Z = 1/\int_\theta \mathcal{P}_\theta(s'|s,a)b(\theta|h)d\theta = 1/\int_\theta \mathcal{P}_\theta(s'|s,a)\tilde{b}(\theta|h)d\theta = 1/\int_\theta \mathcal{P}_\theta(s'|s,a)\tilde{b}(\theta|ha)d\theta$ is the normalization constant. The fifth equality in Eq. (6) holds due to the inductive hypothesis. The sixth equality is based on the fact that the choice of $a$ at each node $h$ is made independently of the sample $\theta$. As for the last equality, to experience $\theta$ at $has'$, the sample $\theta$ needs to traverse $ha$ (with probability $\tilde{b}(\theta|ha)$) and then the state $s'$ needs to be sampled, which is with probability $\mathcal{P}_\theta(s'|s,a)$, so $\tilde{b}(\theta|has') \propto \tilde{b}(\theta|ha)\mathcal{P}_\theta(s'|s,a)$. □

# B  CONSISTENCY PROOF OF CONTINUOUS BAMCP

This appendix provides the detailed proof for Theorem 4.1. Our proof adapts the consistency proof methodology developed for Polynomial Upper Confidence Trees (PUCT) (Auger et al., 2013). To apply this framework, we first note that the Pessimistic Bayes-Adaptive MDP (P-BAMDP) defined in the main text can be formally treated as a standard, albeit large-state-space, MDP. **Throughout this proof, we will refer to this mathematical object as the Pessimistic MDP (P-MDP)**. Our goal is to show that Continuous BAMCP is a consistent solver for this P-MDP.

The remainder of this appendix is organized as follows. Section B.1 formally defines the P-MDP and introduces the key notations, assumptions, and the induction property central to our proof. Section B.2 establishes the base case for our backward induction. Sections B.3 and B.4 present the two core inductive steps, showing how the consistency property propagates upward through the random and decision nodes of the search tree, respectively. Finally, Section B.5 concludes the proof.

## B.1  PREAMBLE AND DEFINITIONS

**P-MDP Formalization**   The P-MDP that our planner solves is defined by the following components:

- **State Space** $\mathcal{S}^+$: The augmented information state space, where each state is $(s, h)$.
- **Action Space** $\mathcal{A}$: The action space of the original problem.
- **Transition Function** $\mathcal{P}^+$: The belief-weighted transition function over information states.
- **Reward Function** $\tilde{r}$: The fixed, bounded, penalized reward function from Eq. (5)

**Relevant Notations**   We adopt the following notations:

- $z$: A decision node, corresponding to an information state $(s, h)$.
- $w$: A random node, corresponding to a state-action pair $(z, a)$.
- $V_p^*(z), V_p^*(w)$: The optimal Bellman value of node $z$ and $w$ under the P-MDP.
- $n(z), n(w)$: The total number of visits to node $z$ and $w$ during MCTS.
- $\hat{V}(z), \hat{V}(w)$: The empirical average return obtained at node $z$ and $w$.
- $V(z)$: The value of a node $z$ estimated by the learned value function from the previous policy iteration.
- $d$: The depth of a node in the search tree, integer for decision node while semi-integer for random node

**Definition B.1** (**Exponentially sure in** $n$). *We say that some property* $(P)$ *depending on an integer* $n$ *is exponentially sure in* $n$ *(denoted e.s.) if there exists positive constant* $C, h$ *such that the probability that* $(P)$ *holds is at least*

$$1 - C\exp(-hn).$$

Our proof relies on the following assumptions:

**Assumption B.2** (**Policy Regularity**). *For any decision node* $z = (s, h)$ *and any precision* $\Delta > 0$, *the policy* $\pi(\cdot|z)$ *used to sample new actions has a non-zero probability of sampling a* $\Delta$-*optimal action within the current P-MDP. Formally, there exist fixed constants* $\theta > 0$ *and* $p > 1$ *such that,*

$$V_p^*(w = (z, a)) \geq V_p^*(z) - \Delta \text{ with probability at least } \min(1, \theta\Delta^p) \tag{7}$$

**Assumption B.3** (**Bounded Value Function Error**). *In our algorithm, when a Monte Carlo Tree Search simulation reaches a leaf node $z$ of the current search tree (i.e., a node that has not yet been expanded), we use a learned value function, $V$, to estimate its value for the backup step. This function is trained as part of the overall policy iteration framework. We assume that $V$ is an imperfect approximator of the P-MDP's true optimal value function, $V_p^*$, and that its approximation error is uniformly bounded by a constant $\epsilon_V \geq 0$ over the entire state space:*

$$|V(z) - V_p^*(z)| \leq \epsilon_V, \quad \forall z \in \mathcal{S}^+$$

*Furthermore, we assume the output of $V(z)$ is bounded.*

The proof aims to establish the following property via backward induction:

**Definition B.4** (**Induction Property Cons($\gamma, f, d$)**). *There exists a constant $C_d > 0$ and a function $f_d(\cdot)$ such that for all decision node $z$ at depth $d$,*

$$|\hat{V}(z) - V_p^*(z)| \leq C_d n(z)^{-\gamma_d} + f_d(\epsilon_V) \text{ e.s. in } n(z)$$

*and for all nodes $w$ at semi-integer depth $d + \frac{1}{2}$,*

$$|\hat{V}(w) - V_p^*(w)| \leq C_{d+\frac{1}{2}} n(w)^{-\gamma_{d+\frac{1}{2}}} + f_{d+\frac{1}{2}}(\epsilon_V) \text{ e.s. in } n(w)$$

*where $\epsilon_V$ is the base approximation error of the value function (from Assumption B.3).*

## B.2 BASE STEP

The backward induction begins at the maximum search depth, $d_{max}$. We aim to establish the induction property **Cons($\gamma, f, d$)** for random nodes $w$ at depth $d = d_{max} - \frac{1}{2}$. As our algorithm uses a learned value function, $V(z_i)$, to evaluate leaf nodes $z_i$ (children of $w$), this step relies on the value function error bound established in Assumption B.3.

Our goal is to bound the total error $|\hat{V}(w) - V_p^*(w)|$. Using the triangle inequality, we decompose this error:

$$|\hat{V}(w) - V_p^*(w)| \leq |\hat{V}(w) - \mathbb{E}[V(z_i)]| + |\mathbb{E}[V(z_i)] - V_p^*(w)|$$

We bound the two terms on the right-hand side separately.

**(a) Bounding the Sampling Error**  The first term, $|\hat{V}(w) - \mathbb{E}[V(z_i)]|$, is the concentration error of the MCTS sampling process. Since the value function output $V(z_i)$ is bounded (per Assumption B.3), we can apply Hoeffding's inequality. Following the original PUCT proof (Auger et al., 2013), we set the deviation tolerance $t = n(w)^{-\frac{1}{3}}$, which gives us the following bound:

$$|\hat{V}(w) - \mathbb{E}[V(z_i)]| \leq O(n(w)^{-\frac{1}{3}}) \text{ e.s. in } n(w)$$

**(b) Bounding the Approximation Error**  The second term, $|\mathbb{E}[V(z_i)] - V_p^*(w)|$, represents the inherent bias from using an approximate value function. We bound it using Assumption B.3:

$$\begin{aligned}
|\mathbb{E}[V(z_i)] - V_p^*(w)| &= |\mathbb{E}[V(z_i)] - \mathbb{E}[V_p^*(z_i)]| \\
&\leq \mathbb{E}[|V(z_i) - V_p^*(z_i)|] \\
&\leq \mathbb{E}[\epsilon_V] = \epsilon_V
\end{aligned}$$

**Conclusion for the Base Step**  By combining the bounds for the sampling error and the approximation error, we obtain the bound on the total error:

$$|\hat{V}(w) - V_p^*(w)| \leq O(n(w)^{-\frac{1}{3}}) + \epsilon_V \text{ e.s. in } n(w)$$

This result precisely matches the form required by our revised induction property, **Cons($\gamma, f, d$)**. We have successfully shown that the property holds for the base case at depth $d = d_{max} - \frac{1}{2}$, with parameters:

- $\gamma_{d_{max}-\frac{1}{2}} = \frac{1}{3}$
- $f_{d_{max}-\frac{1}{2}}(\epsilon_V) = \epsilon_V$

## B.3 INDUCTIVE STEP: FROM DECISION TO RANDOM NODES

In this section, we perform the first inductive step, moving from decision nodes up to random nodes. We assume the induction property **Cons**$(\gamma, f, d)$ holds for all decision nodes $z$ at depth $d$. Our objective is to prove that, as a consequence, the property also holds for their parent nodes, which are the random nodes $w$ at depth $d - \frac{1}{2}$.

A key distinction in our algorithm is the special handling of the very first action sampled from a decision node. This requires us to categorize all random nodes $w$ (at depth $d - \frac{1}{2}$) based on their history:

1. **Category 1: First-Child Nodes** ($w_0$): A random node that was the first child ever created from its parent decision node. Its initial value estimate was derived directly from our learned value function, $V$.

2. **Category 2: Standard Nodes** ($w_j, j > 0$): All other random nodes. Their value estimates have always been computed through the standard MCTS backup process.

We will analyze these two categories separately to show that the induction property holds universally.

### B.3.1 ANALYSIS OF STANDARD RANDOM NODES ($w_j, j > 0$)

For a standard random node $w$ in this category, the proof structure closely follows Section 5 of Auger et al. (2013). We aim to bound the total error $|\hat{V}(w) - V_p^*(w)|$.

Let $i_0 = \lfloor n(w)^\alpha \rfloor$ be the newest child of $w$ (a decision node at depth $d$), and let $r = i_0 - 1$ be the number of "mature" children.(i.e., all children except for the newest). Let $R_w(i) = r_i + \gamma \hat{V}(i)$ denote the return through child $i$. We decompose the total error into four terms:

$$|\hat{V}(w) - V_p^*(w)| \leq \underbrace{|\sum_{i=1}^{r} \left(\frac{n(i)}{n(w)} - \frac{1}{r}\right) R_w(i)|}_{\text{(a) Weight Deviation Error}} + \underbrace{|\frac{1}{r} \sum_{i=1}^{r} (R_w(i) - \mathbb{E}[R_w(j)])|}_{\text{(b) Sampling Error of "Mature" Returns}}$$

$$+ \underbrace{|\mathbb{E}[R_w(j)] - V_p^*(w)|}_{\text{(c) Bias / Approximation Error}} + \underbrace{|\frac{n(i_0)}{n(w)} R_w(i_0)|}_{\text{(d) "Newest" Child Error}}$$

We now bound each term:

**(a) Bounding the Weight Deviation Error**  This term is bounded by leveraging the fact that the children of a random node are explored in a nearly uniform manner (a consequence of the DPW mechanism, as shown in Lemma 1 of Auger et al. (2013). Following their derivation, this term is bounded by $O(n(w)^{-(1-\alpha)} + n(w)^{-\alpha})$.

**(b) Bounding the Sampling Error of "Mature" Returns**  This term represents the concentration error for the returns of the mature child nodes. The returns $R_w(i)$ are i.i.d. bounded random variables. Therefore, we can apply Hoeffding's inequality. To align with the proof by Auger et al. (2013) and derive the resulting convergence rate for this layer, we select the deviation tolerance $t$ as:

$$t := n(w)^{-\frac{\gamma_d}{1+3\gamma_d}}$$

With this choice, the application of Hoeffding's inequality shows that the error for this term is bounded by $O(n(w)^{-\frac{\gamma_d}{1+3\gamma_d}})$ and that this bound holds exponentially surely.

**(c) Bounding the Bias / Approximation Error**  This term is unique to our proof and requires our induction hypothesis.

$$\begin{aligned}|\mathbb{E}[R_w(j)] - V_p^*(w)| &= |\mathbb{E}[r_j + \gamma \hat{V}(j)] - \mathbb{E}[r_j + \gamma V_p^*(j)]| \\ &= \gamma|\mathbb{E}[\hat{V}(j) - V_p^*(j)]| \\ &\leq \gamma \mathbb{E}[|\hat{V}(j) - V_p^*(j)|]\end{aligned}$$

By the induction hypothesis for nodes at depth $d$, we have $|\hat{V}(j) - V_p^*(j)| \leq C_d n(j)^{-\gamma_d} + f_d(\epsilon_V)$. Substituting this into the inequality gives:

$$|\mathbb{E}[R_w(j)] - V_p^*(w)| \leq \gamma \mathbb{E}[C_d n(j)^{-\gamma_d}] + \gamma f_d(\epsilon_V) \text{ e.s. in } n(w)$$

The first part of this bound is a decaying term, while the second part shows how the bias propagates.

**(d) Bounding the "Newest" Child Error**   Following the same logic as Auger et al. (2013), this term is bounded by $O(n(w)^{-\alpha})$ because the newest child has been visited only a small number of times.

**Conclusion for Standard Nodes**   Combining these bounds, the overall convergence rate is determined by the slowest-converging term. Following the derivation in the PUCT paper, setting the exploration parameter $\alpha = \frac{3\gamma_d}{1+3\gamma_d}$, establishes the slowest convergence rate as $\frac{\gamma_d}{1+3\gamma_d}$. The bias term is propagated from the layer below. Thus, for any standard random node, the **Cons($\gamma, f, d$)** property holds.

### B.3.2 ANALYSIS OF THE FIRST-CHILD RANDOM NODE ($w_0$)

For the first child $w_0$, the value estimate includes a special term from its initial visit. On the first visit, the algorithm samples a next state, which we denote $s_0'$, and receives a corresponding reward $r_0$. The next state's value is thus initially estimated using the learned function, $V(s_0')$. The error analysis for this random node, $|\hat{V}(w_0) - V_p^*(w_0)|$, will contain the same four error terms as in the standard case (for the subsequent backups), plus an additional error term originating from the first visit:

$$\text{Additional Error} = \left| \frac{1}{n(w_0)}(r_0 + \gamma V(s_0')) - \frac{1}{n(w_0)} V_p^*(w_0) \right|$$

Since $r_0$, $V(s_0')$, and $V_p^*(w_0)$ are all bounded constants, this additional error term is bounded by $O(n(w_0)^{-1})$.

The overall convergence rate for $w_0$ is determined by the minimum of the rates of all error components. From the PUCT derivation, we know that $\gamma_d \leq 1$, which implies $\gamma_{d-1/2} = \frac{\gamma_d}{1+3\gamma_d} < 1$. Therefore, the convergence rate of the additional error term, $O(n(w_0)^{-1})$, is faster than the bottleneck rate of the standard MCTS error terms, $O(n(w_0)^{-\gamma_{d-1/2}})$. This special term does not change the overall convergence rate. Thus, the **Cons($\gamma, f, d$)** property also holds for the first-child random node $w_0$ with the exact same parameters.

### B.3.3 CONCLUSION FOR THIS SECTION

We have shown that for any random node $w$ at depth $d - \frac{1}{2}$, if its children at depth $d$ satisfy the **Cons($\gamma, f, d$)** property, then $w$ itself also satisfies the property. The parameters of the property propagate upward according to the following recursive formulas:

- $\gamma_{d-\frac{1}{2}} = \frac{\gamma_d}{1+3\gamma_d}$
- $f_{d-\frac{1}{2}}(\epsilon_V) = \gamma f_d(\epsilon_V)$

## B.4   INDUCTIVE STEP: FROM RANDOM TO DECISION NODES

In this section, we perform the second inductive step.   We assume the induction property **Cons($\gamma, f, d$)** holds for all random nodes $w$ at depth $d + \frac{1}{2}$. Our objective is to prove that the property also holds for their parent nodes, which are the decision nodes $z$ at depth $d$. The proof adapts the structure of Section 6 in Auger et al. (2013), carefully accounting for the bias term $f_{d+\frac{1}{2}}(\epsilon_V)$ from our induction hypothesis. First we establish an upper bound on $\hat{V}(z) - V_p^*(z)$, and then a lower bound.

### B.4.1 UPPER BOUND ON THE ERROR

To establish an upper bound on $\hat{V}(z) - V_p^*(z)$, we partition the children $w_i$ of $z$ into two classes, with a parameter $\epsilon < 1 - \alpha$ to be determined later:

- **Class I**: Children $w_i$ with few visits, $n(w_i) \leq n(z)^{1-\alpha-\epsilon}$.
- **Class II**: All other children, for which $n(w_i) > n(z)^{1-\alpha-\epsilon}$.

For Class I, the number of children is at most $n(z)^{\alpha}$, so their total contribution is bounded by $\frac{n(z)^{\alpha} \cdot n(z)^{1-\alpha-\epsilon}}{n(z)} = n(z)^{-\epsilon}$. For Class II, we apply the induction hypothesis for nodes at depth $d + \frac{1}{2}$:

$$
\begin{aligned}
\sum_{i \in \text{Class II}} \frac{n(w_i)}{n(z)}(\hat{V}(w_i) - V_p^*(z)) &\leq \sum_{i \in \text{Class II}} \frac{n(w_i)}{n(z)}(\hat{V}(w_i) - V_p^*(w_i)) \\
&\leq \sum_{i \in \text{Class II}} \frac{n(w_i)}{n(z)}(C_{d+\frac{1}{2}} n(w_i)^{-\gamma_{d+\frac{1}{2}}} + f_{d+\frac{1}{2}}(\epsilon_V)) \\
&\leq C_{d+\frac{1}{2}} \left(n(z)^{1-\alpha-\epsilon}\right)^{-\gamma_{d+\frac{1}{2}}} + f_{d+\frac{1}{2}}(\epsilon_V)
\end{aligned}
$$

Combining the two parts, we get:

$$
\hat{V}(z) - V_p^*(z) \leq n(z)^{-\epsilon} + C_{d+\frac{1}{2}} n(z)^{-\gamma_{d+\frac{1}{2}}(1-\alpha-\epsilon)} + f_{d+\frac{1}{2}}(\epsilon_V)
$$

To optimize this bound, we balance the exponents by choosing $\epsilon = \frac{\gamma_{d+\frac{1}{2}}(1-\alpha)}{1+\gamma_{d+\frac{1}{2}}}$. This leads to the final upper bound for the error:

$$
\hat{V}(z) - V_p^*(z) \leq (1 + C_{d+\frac{1}{2}})n(z)^{-\frac{\gamma_{d+\frac{1}{2}}(1-\alpha)}{1+\gamma_{d+\frac{1}{2}}}} + f_{d+\frac{1}{2}}(\epsilon_V)
$$

### B.4.2 LOWER BOUND ON THE ERROR

The proof for the lower bound is more involved. It relies on the parameter definitions from the PUCT paper, as well as two key results regarding the UCT exploration mechanism, which we state here for completeness before proceeding.

**Lemma B.5** (Adapted from Lemma 3, Auger et al. (2013)). *Consider a bandit setting where the score of a child node $w_i$ at time $n$ is computed by the UCT rule:*

$$
sc_n(w_i) = \hat{V}_n(w_i) + \sqrt{\frac{g(n)}{n(w_i)}}
$$

*where $g(n)$ is a non-decreasing exploration function. If $i$ denotes the $i^{th}$ constucted child, for all $n > i^{\frac{1}{\alpha(1-\alpha)}}$ we have*

$$
n(w_i) \geq \frac{1}{4} \min\left(g(n^{1-\alpha}), n^{1-\alpha}\right).
$$

**Corollary B.6** (Adapted from Corollary 3, Auger et al. (2013)). *For the exploration function $g(n) = n^e$ with $0 < e < 1$, we obtain*

$$
n(w_i) \geq \frac{1}{4} n^{e(1-\alpha)} \text{ if } i \leq n^{\alpha(1-\alpha)}. \tag{8}
$$

With these results, we can define the parameters as functions of $\gamma_{d+\frac{1}{2}}$ and $p$ (from Assumption B.2) for this layer and proceed with the three-step proof structure.

- Progressive widening coefficient: $\alpha := \frac{\gamma_{d+\frac{1}{2}}}{1+4\gamma_{d+\frac{1}{2}}}$

- Exploration coefficient: $e := \frac{1}{2p\left(1+4\gamma_{d+\frac{1}{2}}\right)}$

- Auxiliary proof parameters: $\xi := \frac{1}{1+e\gamma_{d+\frac{1}{2}}(1-\alpha)}$ and $\Delta := \left(\frac{1}{4}n(z)^{\xi e(1-\alpha)}\right)^{-\gamma_{d+\frac{1}{2}}}$

**Step 1: Existence of a near-optimal child**   Based on Assumption B.2 (Policy Regularity), a direct adaptation of the proof in Auger et al. (2013) shows that a near-optimal child node is discovered with high probability. Formally: *Exponentially surely in $n(z)$, there exists at time $\lceil n(z)^{\xi(1-\alpha)} \rceil$ a child $w_0$ of $z$ such that*

$$V_p^*(w_0) \geq V_p^*(z) - \Delta \text{ and } i_0 \leq n(z)^{\xi(1-\alpha)\alpha} \tag{9}$$

**Step 2: Score of children selected at later stages**   The objective of this step is to show that any child selected after time $n(z)^{\xi}$ must have a high score. Let $n'$ be such that $n^{\xi} \leq n' \leq n$. And, by Corollary B.6,

$$n'(w_0) \geq \frac{1}{4} n'^{e(1-\alpha)} \geq \frac{1}{4} n^{\xi e(1-\alpha)}.$$

First, according to Corollary B.6, the near-optimal child $w_0$ discovered in Step 1 will be visited a number of times $n(w_0)$ that grows polynomially with $n(z)$.

We can therefore apply our induction hypothesis **Cons**$(\gamma, f, d)$ to $w_0$, with the Eq.(9), there exists a $C' > 0$ that:

$$\hat{V}(w_0) \geq V_p^*(w_0) - C'\left(\frac{1}{4} n^{\xi e(1-\alpha)}\right)^{-\gamma_{d+\frac{1}{2}}} - f_{d+\frac{1}{2}}(\epsilon_V)$$
$$\geq V_p^*(z) - (1 + C')\Delta - f_{d+\frac{1}{2}}(\epsilon_V)$$

Now, consider any child $w_1$ selected by the UCT rule at a later time $n' \geq n(z)^{\xi}$. Its score must be at least as high as the score of $w_0$. This gives the key inequality:

$$\hat{V}(w_1) + \sqrt{\frac{n'^e}{n'(w_1)}} \geq \hat{V}(w_0) + \sqrt{\frac{n'^e}{n'(w_0)}}$$

Combining these results, we can establish a lower bound on the estimated value $\hat{V}(w_1)$ of any child selected at a late stage of the search.

$$\hat{V}(w_1) + \sqrt{\frac{n}{n'(w_1)}} \geq V_p^*(z) - (1 + C')\Delta - f_{d+\frac{1}{2}}(\epsilon_V)$$

As shown by Auger et al. (2013), this property holds exponentially surely in $n(z)$.

**Step 3: Bounding the value estimate $\hat{V}(z)$**   Consider a child $w_1$ selected after $n(z)^{\xi}$. By the previous step, exponentially in $n(z)$, this child must either satisfy

$$\sqrt{\frac{n^e}{n(w_1)}} \geq \Delta \tag{10}$$

$$\text{or } \hat{V}(w_1) \geq V(z) - (2 + C')\Delta - f_{d+\frac{1}{2}}(\epsilon_V) \tag{11}$$

We now bound the average value $\hat{V}(z)$ by partitioning the children of $z$ into three categories:

1. Children visited only before time $n(z)^{\xi}$.
2. Children visited after $n(z)^{\xi}$ satisfying (10)
3. Children visited after $n(z)^{\xi}$ satisfying (11).

We bound the contribution of each category to the total error sum. The contribution from **Category 1** is bounded by $O(n(z)^{\xi-1})$, since we have

$$\left| \sum_{w \text{ in cat.1}} \frac{n(w)}{n(z)} \left( \hat{V}(w) - V_p^*(z) \right) \right| \leq \frac{\sum_{w \text{ in cat.1}} n(w)}{n(z)} \leq \frac{n(z)^{\xi}}{n(z)}$$

For **Category 2**, the definition of the category implies $n(w_i) \leq n(z)^e/\Delta^2$. Since there are at most $n(z)^\alpha$ children, their total contribution to the error sum is bounded by $O(n(z)^{\alpha+e-1}/\Delta^2)$. For **Category 3**, the bound on $\hat{V}(w_i)$ from Step 2 introduces the crucial $-f_{d+1/2}(\epsilon_V)$ term.

Combining the bounds from all three categories gives the final lower bound on the error. Following the argument in Auger et al. (2013):

$$\hat{V}(z) - V_p^*(z) \geq \underbrace{-(2+C')\Delta(1-n(z)^{\xi-1}) - f_{d+\frac{1}{2}}(\epsilon_V)}_{\text{cat.3}} - \underbrace{n(z)^{\xi-1}}_{\text{cat.1}} - \underbrace{\frac{n(z)^{\alpha+e-1}}{\Delta^2}}_{\text{cat.2}}$$

By comparing the magnitudes of the decaying terms with the chosen parameter definitions, as is done in Auger et al. (2013), this complex expression simplifies to the desired form.

### B.4.3 Conclusion for this Section

By combining the results from the upper and lower bound analyses, we conclude that the total error at a decision node $z$ is bounded as:

$$|\hat{V}(z) - V_p^*(z)| \leq C_d n(z)^{-\gamma_d} + f_{d+\frac{1}{2}}(\epsilon_V)$$

This demonstrates that the induction property **Cons**$(\gamma, f, d)$ holds for depth $d$. The parameters propagate upward according to the following recursive formulas, which are identical to those derived by Auger et al. (2013):

- $C_d = (5 + C')4^{\gamma_{d+\frac{1}{2}}}$
- $\gamma_d = \frac{\gamma_{d+\frac{1}{2}}}{1+7\gamma_{d+\frac{1}{2}}}$
- $f_d(\epsilon_V) = f_{d+\frac{1}{2}}(\epsilon_V)$

### B.5 Conclusion of the proof

In the preceding sections, we have established all the necessary components for our backward induction argument.

- In Section B.2, we established the base case, proving that the induction property **Cons**$(\gamma, f, d)$ holds for random nodes at depth $d = d_{max} - \frac{1}{2}$, with initial parameters $\gamma_{d_{max}-\frac{1}{2}} = \frac{1}{3}$ and $f_{d_{max}-\frac{1}{2}}(\epsilon_V) = \epsilon_V$.
- In Section B.3, we proved the inductive step from decision nodes up to random nodes, deriving the recursions $\gamma_{d-\frac{1}{2}} = \frac{\gamma_d}{1+3\gamma_d}$ and $f_{d-\frac{1}{2}}(\epsilon_V) = \gamma f_d(\epsilon_V)$.
- In Section B.4, we proved the inductive step from random nodes up to decision nodes, deriving the recursions $\gamma_d = \frac{\gamma_{d+\frac{1}{2}}}{1+7\gamma_{d+\frac{1}{2}}}$ and $f_d(\epsilon_V) = f_{d+\frac{1}{2}}(\epsilon_V)$.

With the base case proven and both inductive steps validated, we conclude by backward induction that the property **Cons**$(\gamma, f, d)$ holds for all depths up to the root of the search tree ($d = 0$).

By solving the recursion for the bias term, we can find the accumulated error at the root. The base error at depth $d_{max} - \frac{1}{2}$ is $f_{d_{max}-\frac{1}{2}}(\epsilon_V) = \epsilon_V$. The error term is multiplied by a discount factor $\gamma$ each time it propagates from a decision node layer up to a random node layer. This occurs $d_{max}$ times from the leaves to the root. Therefore, the bias term at the root node ($d = 0$) is $f_0(\epsilon_V) = \gamma^{d_{max}}\epsilon_V$.

Specifically, for the root node $z_0$, its total error is bounded by:

$$|\hat{V}(z_0) - V_p^*(z_0)| \leq C_0 n(z_0)^{-\gamma_0} + \gamma^{d_{max}}\epsilon_V$$

where $\gamma_0 > 0$ and $C_0$ are constants derived from the recursions. This shows that as the number of simulations $E = n(z_0) \rightarrow \infty$, the estimated value of the root node converges to a ball of radius

$\gamma^{d_{max}} \epsilon_V$ around the true optimal value of the P-MDP. Notably, the size of this error ball, which stems from our function approximator, shrinks exponentially with the search depth $d_{max}$.

This completes the proof of Theorem 4.1, establishing that Continuous BAMCP is a consistent planner for the P-MDP. $\qquad\square$

## C    ALTERNATIVE DESIGN CHOICES FOR CONTINUOUS BAMCP

The terms labeled in blue in Algorithm 1 have alternative design choices, which are introduced as below. Empirical comparisons among these alternatives are reserved for future work.

$\tilde{Q}((s,h),x)$ in the exploration strategy, i.e., $a \leftarrow \arg\max_{x \in C((s,h))} \tilde{Q}((s,h),x)$, could take various forms. For instance, in Couëtoux et al. (2011); Guez et al. (2013); Lee et al. (2020), $\tilde{Q}((s,h),x) = Q((s,h),x) + c\sqrt{\frac{\log N((s,h))}{N((s,h),x)}}$; in PUCT (Auger et al., 2013), $\tilde{Q}((s,h),x) = Q((s,h),x) + \sqrt{\frac{N((s,h))^{e(d)}}{N((s,h),x)}}$, where $e(d)$ is a schedule of coefficients related to the search depth $d$; in Sampled MuZero (a variant of MuZero that can be applied in continuous action spaces (Hubert et al., 2021)), $\tilde{Q}((s,h),x) = Q((s,h),x) + \hat{\pi}(x|(s,h))\frac{\sqrt{N((s,h))}}{1+N((s,h),x)}\left(c_1 + \log\left(\frac{N((s,h))+c_2+1}{c_2}\right)\right)$. Here, $c$, $c_1$, $c_2$ are hyperparameters, $\hat{\pi} = \hat{\beta}\pi^{1-1/\tau}$ is a sample policy defined upon the real policy $\pi$. In particular, at each decision point $(s,h)$, Sampled MuZero would sample $M$ actions $\{a_i\}$ from the distribution $\pi^{1/\tau}$ and accordingly define $\hat{\beta}(a|(s,h)) = \sum_i \mathbb{1}_{a_i=a}/M$, where $\tau > 0$ is a temperature hyperparameter. **Thus, Sampled MuZero does not adopt progressive widening like ours.** Following BAMCP, we adopt the first definition of $\tilde{Q}((s,h),x)$, though it could potentially be improved with other choices. In addition, as an implementation trick (Hamrick et al., 2021), the Q estimates are usually normalized into $\bar{Q} \in [0,1]$ before being used to calculate $\tilde{Q}$ as above. The normalized estimates can be computed as $\bar{Q}((s,h),x) = \frac{Q((s,h),x)-Q_{\min}}{Q_{\max}-Q_{\min}}$, where $Q_{\max}$ and $Q_{\min}$ are the maximum and minimum Q values observed in the search tree so far.

As the planning/search result, $\pi_{\text{ret}}$ can take multiple forms. In Guez et al. (2013); Sunberg & Kochenderfer (2018); Lee et al. (2020), $\pi_{\text{ret}}((s,h)) = \arg\max_{a \in C((s,h))} Q((s,h),a)$; in (Sampled) MuZero, $\pi_{\text{ret}}(a|(s,h)) = \frac{N((s,h),a)^{1/\tau}}{\sum_{x \in C((s,h))} N((s,h),x)^{1/\tau}}$; in ROSMO (a variant of MuZero with improved performance in offline scenarios (Liu et al., 2023)), $\pi_{\text{ret}}(a|(s,h)) \propto \pi(a|(s,h))\exp(Q((s,h),a) - V((s,h)))$. Here, $\tau \in (0,1]$ is a temperature parameter and decays with the training process, ensuring the action selection becomes greedier. We select the second form for $\pi_{\text{ret}}$ in Algorithm 1. This is because (1) as described in PUCT, the returned action should be the most visited one, which is not necessarily the one with the highest Q value, and (2) ROSMO adopts one-step look-ahead rather than deep tree search at each root node, which does not align with our approach.

As for the conditions of double progressive widening, PUCT designs $\alpha$ and $\beta$ to be functions of the search depth $d$, while UCT-DPW (Couëtoux et al., 2011) utilizes a different set of conditions: $\lceil K_a N((s,h))^\alpha \rceil \geq |C((s,h))|$, $\lceil K_s N((s,h),a)^\beta \rceil \geq |C((s,h),a)|$, where $K_a$, $K_s$, $\alpha$, $\beta$ are all constant hyperparameters. Moreover, when the progressive widening condition for sampling the next state is not satisfied, either the least visited node in $C((s,h),a)$ can be selected (following PUCT), or a random node can be sampled from $C((s,h),a)$ according to a distribution proportional to the number of visits (following UCT-DPW). As shown in Algorithm 1, **we follow the designs of PUCT**, but keep $\alpha$ and $\beta$ as constants for simplicity in hyperparameter fine-tuning.

Finally, the condition for continuing the simulation procedure, i.e., $N((s,h)) > 1$, could potentially be replaced with $N((s,h),a) > 1$ or $N((s',hars')) > 0$. These conditions indicate that the nodes $(s,h)$, $(s,h,a)$, and $(s',hars')$ have been visited before, respectively. At the end of the simulation procedure, we can either apply rollouts, i.e., simulating a single path until the end of an episode, to estimate the expected value for a leaf node $(s,h)$, or directly use $V((s,h))$ as the estimation. The former approach is widely used in online planning algorithms (Guez et al., 2013; Sunberg & Kochenderfer, 2018; Lee et al., 2020), while the latter is used in policy iteration frameworks like ours.

# D KEY HYPERPARAMETER SETUP

| Data Type | Environment | BA-MBRL | | | | BA-MCTS | | | | BA-MCTS-SL | | | |
|---|---|---|---|---|---|---|---|---|---|---|---|---|---|
| | | $K$ | $\lambda$ | $H$ | $N$ | $K$ | $\lambda$ | $H$ | $N$ | $K$ | $\lambda$ | $H$ | $N$ |
| random | HalfCheetah | 10 | 7 | 6 | 200 | 10 | 7 | 6 | 800 | 10 | 7 | 6 | 500 |
| random | Hopper | 6 | 50 | 47 | 700 | 6 | 50 | 47 | 800 | 6 | 50 | 47 | 500 |
| random | Walker2d | 10 | 0.5 | 20 | 700 | 10 | 0.5 | 20 | 800 | 10 | 0.5 | 20 | 500 |
| medium | HalfCheetah | 12 | 6 | 6 | 300 | 12 | 6 | 5 | 800 | 12 | 6 | 5 | 500 |
| medium | Hopper | 12 | 40 | 42 | 200 | 12 | 40 | 42 | 800 | 12 | 40 | 42 | 200 |
| medium | Walker2d | 8 | 5 | 20 | 700 | 8 | 5 | 20 | 800 | 8 | 5 | 20 | 500 |
| med-replay | HalfCheetah | 11 | 40 | 10 | 300 | 11 | 40 | 10 | 800 | 11 | 40 | 10 | 500 |
| med-replay | Hopper | 7 | 5 | 5 | 700 | 7 | 5 | 5 | 800 | 7 | 5 | 5 | 500 |
| med-replay | Walker2d | 13 | 2.5 | 47 | 1000 | 13 | 2.5 | 47 | 800 | 13 | 2.5 | 47 | 500 |
| med-expert | HalfCheetah | 7 | 100 | 5 | 1000 | 7 | 100 | 5 | 800 | 7 | 100 | 5 | 1100 |
| med-expert | Hopper | 12 | 40 | 43 | 600 | 12 | 40 | 43 | 800 | 12 | 40 | 43 | 500 |
| med-expert | Walker2d | 6 | 20 | 37 | 400 | 6 | 20 | 37 | 800 | 6 | 20 | 37 | 500 |

Table 5: Key hyperparameters of the proposed algorithms for each evaluation task in the ablation study. $K$: ensemble size, $\lambda$: reward penalty coefficient, $H$: rollout horizon, $N$: number of training epochs.

Table 5 lists the key hyperparameters of BA-MCTS and its ablated variants used in the ablation study. **For each task, an ensemble of $K$ dynamics and reward models is trained using the provided offline dataset. These learned models are then utilized as a simulator to train a control policy using off-the-shelf RL methods, such as SAC. The policy is trained for $N$ epochs. At each epoch, $50000H$ transitions are sampled by interacting with the simulator, followed by $1000$ RL training iterations. In particular, $50000$ states are randomly sampled from the offline dataset, with each state followed by a rollout lasting $H$ time steps. To mitigate overestimation, a reward penalty based on the discrepancy among the ensemble members is applied with a coefficient $\lambda$, as shown in Equation (5).** The setups for $K$, $\lambda$, and $H$ are almost the same across the three algorithms and primarily inherited from the baseline – "Optimized" (Lu et al., 2022), to make sure the improvements are brought by the Bayesian RL and deep search components (rather than tricky hyperparameter setups).

The policy is evaluated on the ground truth environment for 10 episodes at the end of each training epoch. For all tables involving benchmarking results on D4RL, we report the average scores across the final 10 training epochs of our algorithms. It is important to note that increasing the number of training epochs $N$ does not necessarily lead to better policy performance, since the training is based on learned dynamics and reward models rather than the ground truth. According to Lu et al. (2022) and our experiments, the hyperparameters listed above can significantly influence the performance of model-based RL. Adjusting these hyperparameters could either enhance or impair the learning performance of our algorithms. We also suspect that the performance of the baselines listed in Table 9, which are from their original papers, could be further improved by fine-tuning the relevant hyperparameters.

BA-MCTS and BA-MCTS-SL utilize Continuous BAMCP (i.e., Algorithm 1) to collect samples for subsequent policy distillation. **Instead of performing a tree search at every state, we randomly select a proportion (i.e., $\rho$) of states from the available 50000 states at each rollout time step as root nodes for tree search to decide on the optimal actions. For the remaining states, actions are sampled directly from the policy, i.e., $a \sim \pi(\cdot|s)$.** The tree search procedure is detailed in Algorithm 1, with the number of MCTS iterations, $E$, set to 50. **Increasing $\rho$ and $E$ can potentially enhance performance, but it will also linearly increase the computational cost.** Table 6 outlines the key hyperparameters related to the search process for each algorithm and task. (1) As described in Algorithm 1, the parameters $\alpha$ and $\beta$ control the rate of double progressive widening. ($\beta$ is set as 0.5 across all tasks.) To encourage deeper search, we limit the number of actions sampled from a state under $n_a$ and the number of next states sampled from an action under $n_s$, respectively. Action selection follows the UCT rule, as discussed in Appendix A, where $c > 0$ balances the exploration and exploitation. Additionally, inspired by the success of MuZero in enhancing exploration, we introduce Dirichlet noise $x_d$ at the root nodes, where actions are sampled from a mixture of distributions: $a \sim \eta x_d + (1 - \eta)\pi(\cdot|s)$

| Data Type | Environment | BA-MCTS | | | | | | BA-MCTS-SL | | | | | | | |
|---|---|---|---|---|---|---|---|---|---|---|---|---|---|---|---|
| | | $\rho$ | $\alpha$ | $c$ | $\eta$ | $n_s$ | $n_a$ | $\rho$ | $\alpha$ | $c$ | $\eta$ | $n_s$ | $n_a$ | $N_{SL}$ | $N_P$ |
| random | HalfCheetah | 0.1 | 0.5 | 2.5 | 0.3 | 1 | 20 | 0.1 | 0.5 | 2.5 | 0.3 | 1 | 20 | 5 | 0 |
| random | Hopper | 0.1 | 0.5 | 2.5 | 0.3 | 1 | 20 | 0.1 | 0.5 | 2.5 | 0.3 | 1 | 20 | 5 | 0 |
| random | Walker2d | 0.1 | 0.5 | 2.5 | 0.3 | 1 | 20 | 0.1 | 0.5 | 2.5 | 0.3 | 1 | 20 | 5 | 100 |
| medium | HalfCheetah | 0.1 | 0.5 | 1.0 | 0.1 | 5 | 10 | 0.1 | 0.5 | 1.0 | 0.1 | 5 | 10 | 20 | 0 |
| medium | Hopper | 0.1 | 0.5 | 2.5 | 0.3 | 1 | 20 | 0.1 | 0.5 | 1.0 | 0.3 | 1 | 20 | 5 | 0 |
| medium | Walker2d | 0.1 | 0.5 | 2.5 | 0.3 | 1 | 20 | 0.1 | 0.5 | 2.5 | 0.3 | 1 | 20 | 5 | 100 |
| med-replay | HalfCheetah | 0.1 | 0.8 | 1.0 | 0.3 | 5 | 10 | 0.1 | 0.8 | 2.5 | 0.3 | 1 | 20 | 5 | 0 |
| med-replay | Hopper | 0.1 | 0.8 | 1.0 | 0.1 | 1 | 20 | 0.1 | 0.8 | 1.0 | 0.1 | 1 | 20 | 15 | 0 |
| med-replay | Walker2d | 0.1 | 0.8 | 2.5 | 0.3 | 1 | 20 | 0.1 | 0.8 | 2.5 | 0.3 | 1 | 20 | 5 | 200 |
| med-expert | HalfCheetah | 0.1 | 0.8 | 1.0 | 0.3 | 5 | 10 | 0.1 | 0.8 | 2.5 | 0.3 | 1 | 20 | 5 | 0 |
| med-expert | Hopper | 0.1 | 0.8 | 1.0 | 0.3 | 1 | 20 | 0.1 | 0.8 | 1.0 | 0.3 | 1 | 20 | 5 | 0 |
| med-expert | Walker2d | 0.1 | 0.8 | 2.5 | 0.3 | 1 | 20 | 0.1 | 0.8 | 2.5 | 0.3 | 1 | 20 | 15 | 100 |

Table 6: Important hyperparameters used in the search process.

and $\eta$ controls the mixture rate. Notably, for $(c, \eta, n_a, n_s)$, we explore the set of possible combinations: $\{(2.5, 0.3, 20, 1), (1.0, 0.3, 20, 1), (1.0, 0.1, 20, 1), (1.0, 0.3, 10, 5), (1.0, 0.1, 10, 5)\}$ during hyperparameter fine-tuning. We believe there are likely more optimal search settings yet to be discovered. (2) In BA-MCTS-SL, policy improvement is achieved through supervised learning. We find that, rather than learning solely from samples collected within the current epoch, incorporating a buffer of samples from the past $N_{SL}$ epochs helps to stabilize the learning process. Also, in the Walker2d environment, BA-MCTS-SL requires a warm-up training phase of $N_P$ epochs using BA-MBRL, allowing the initial policy to generate effective signals for supervised learning.

# E   DETAILS OF THE TOKAMAK CONTROL TASKS

| STATE SPACE | |
|---|---|
| Scalar States | $\beta_n$, Internal Inductance, Line Averaged Density, Loop Voltage, Stored Energy |
| Profile States | Electron Density, Electron Temperature, Pressure, Safety Factor, Ion Temperature, Ion Rotation |

| ACTION SPACE | |
|---|---|
| Targets | Current Target, Density Target |
| Shape Variables | Elongation, Top Triangularity, Bottom Triangularity, Minor Radius, Radius and Vertical Locations of the Plasma Center |
| Direct Actuators | Power Injected, Torque Injected, Total Deuterium Gas Injection, Total ECH Power, Magnitude and Sign of the Toroidal Magnetic Field |

Table 7: The state and action spaces of the tokamak control tasks.

Nuclear fusion is a promising energy source to meet the world's growing demand. It involves fusing the nuclei of two light atoms, such as hydrogen, to form a heavier nucleus, typically helium, releasing energy in the process. The primary challenge of fusion is confining a plasma, i.e., an ionized gas of hydrogen isotopes, while heating it and increasing its pressure to initiate and sustain fusion reactions. The tokamak is one of the most promising confinement devices. It uses magnetic fields acting on hydrogen atoms that have been ionized (given a charge) so that the magnetic fields can exert a force on the moving particles (Pironti & Walker, 2005).

Char et al. (2024) trained a deep recurrent network as a dynamics model for the DIII-D tokamak, a device located in San Diego, California, and operated by General Atomics, using a large dataset of operational data from that device. A typical shot (i.e., episode) on DIII-D lasts around 6-8 seconds, consisting of a one-second ramp-up phase, a multi-second flat-top phase, and a one-second ramp-down phase. The DIII-D also features several real-time and post-shot diagnostics that measure the magnetic equilibrium and plasma parameters with high temporal resolution. **The authors demonstrate that the learned model predicts these measurements for entire shots with remarkable accuracy. Thus, we use this model as a "ground truth" simulator for tokamak control tasks.**

**Specifically, we generate a dataset of 725270 transitions for offline RL and evaluate the learned policy using this data-driven simulator.**

The state and action spaces for the tokamak control tasks are outlined in Table 7. For detailed physical explanations of their components, please refer to Abbate et al. (2021); Char et al. (2023); Ariola et al. (2008). The state space consists of five scalar values and six profiles which are discretized measurements of physical quantities along the minor radius of the toroid. After applying principal component analysis (Maćkiewicz & Ratajczak, 1993), the pressure profile is reduced to two dimensions, while the other profiles are reduced to four dimensions each. In total, the state space comprises 27 dimensions. The action space includes direct control actuators for neutral beam power, torque, gas, ECH power, current, and magnetic field, as well as target values for plasma density and plasma shape, which are managed through a lower-level control module. Altogether, the action space consists of 14 dimensions. While for certain tasks, it is possible to prune the state and action spaces to reduce the learning complexity, we have chosen not to apply any domain-specific knowledge in these evaluations for general RL algorithms. We reserve the domain-specific applications of our algorithms, which would require more domain knowledge and engineering efforts, as an important future work.

We select a reference shot from DIII-D, which spans 251 time steps, and use its trajectories of Ion Rotation, Electron Temperature, and $\beta_n$ as targets for three tracking tasks. Specifically, $\beta_n$ is the normalized ratio between plasma pressure and magnetic pressure, a key quantity serving as a rough economic indicator of efficiency. Since the tracking targets vary over time, we include the time step as part of the policy input. **The reward function for each task is defined as the negative squared tracking error of the corresponding component (i.e., temperature, rotation, or $\beta_n$) at each time step, and the reward is normalized by the episode horizon (i.e., 251 time steps). Notably, for policy learning, the reward function is provided rather than learned from the offline dataset as in D4RL tasks; and the dataset (for offline RL) does not include the reference shot or any nearby, similar shots, making pure imitation infeasible.**

## F    ADDITIONAL EVALUATION RESULTS

This appendix provides detailed empirical results and analyses to supplement the findings presented in Section 5. **Appendix F.1** offers both quantitative and qualitative analyses of the benefits of belief adaptation, presenting additional results for **Section 5.1**.

The following appendices report additional experimental results for benchmarking on MuJoCo D4RL tasks (i.e., **Section 5.2**):

- **Appendix F.2:** Evaluates our algorithm against a suite of model-free offline RL methods.
- **Appendix F.3:** Presents a comprehensive comparison against 19 offline RL baselines, including a statistical significance analysis.
- **Appendix F.4:** Provides a detailed comparison with MuZero and its variants, focusing on their algorithm design, empirical performance, and computational cost.

Finally, we provide complete results for the ablation study (i.e., **Section 5.3**):

- **Appendix F.5:** Demonstrates the significant performance enhancement achieved by the belief adaptation module and our MCTS-based planner.
- **Appendix F.6:** Discusses the trade-offs between the two policy distillation mechanisms investigated (i.e., SAC and supervised learning).
- **Appendix F.7:** Presents an ablation study on the necessity of using reward penalty in offline MBRL.

### F.1    QUANTITATIVE AND QUALITATIVE ANALYSIS OF THE EFFECTIVENESS OF BELIEF ADAPTATION

To understand why belief adaptation over an ensemble of models using a Bayes-Adaptive MDP (BAMDP) is effective, we analyze its impact on the prediction accuracy of the learned world models.

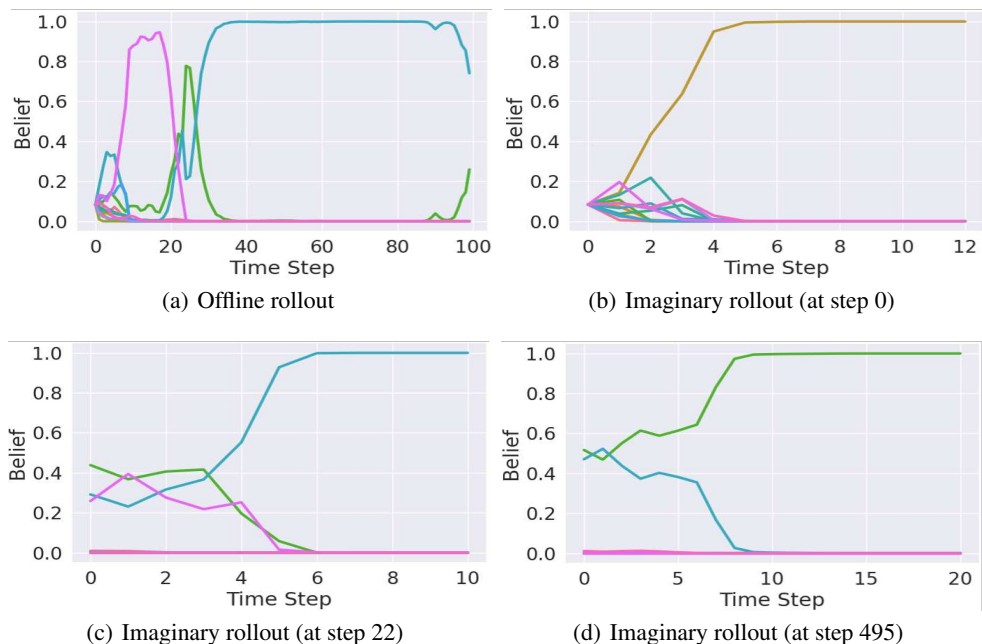

(a) Offline rollout

(b) Imaginary rollout (at step 0)

(c) Imaginary rollout (at step 22)

(d) Imaginary rollout (at step 495)

Figure 1: Belief adaptation during offline and imaginary rollouts. (a) shows the belief over twelve ensemble members, adapting to an offline trajectory of Hopper-med-expert. (b), (c), and (d) illustrate the belief changes during imaginary rollouts which start from different points of the offline trajectory shown in (a).

The Bayesian adaptation, as defined in Eq. (4), uses observed state transition sequences to adjust the belief over each ensemble member, thereby enhancing the quality of predicted rollout trajectories.

For each benchmarking task in D4RL MuJoCo, we compute the average transition likelihood (i.e., $\mathcal{P}(s'|s,a)\mathcal{R}(r|s,a)$) for the provided offline rollouts, comparing cases with and without belief adaptation. The ratios of these likelihoods are presented in Table 8. Results show that the offline rollouts, collected from real environments, are more likely under the adapted ensemble. This demonstrates that Bayesian belief adaptation serves as an effective calibration method for the learned world models.

| hc-med-expert | hc-med-replay | hc-medium | hc-random |
|---|---|---|---|
| 0.7515 | 2.6315 | 2.2674 | 2.4561 |
| hp-med-expert | hp-med-replay | hp-medium | hp-random |
| 14.064 | 4.2214 | 3.5913 | 1.7400 |
| wk-med-expert | wk-med-replay | wk-medium | wk-random |
| 2.0898 | 3.8169 | 34.264 | 1.0341 |

Table 8: Ratios of average transition likelihoods in offline data with and without belief adaptation across ensemble members. A ratio greater than 1 indicates that the adapted ensemble assigns higher likelihood to the true transitions.

Furthermore, we evaluate the quality of imaginary rollouts generated by the learned models. Table 1 in the main content presents the prediction errors for next states and rewards in these rollouts. The results reveal that the adaptive ensemble achieves more accurate predictions in 10 out of 12 benchmarking tasks, indicating that belief adaptation leads to a more accurate model for planning.

As a qualitative analysis, we plot the belief adaptation over an offline trajectory (from $\mathcal{D}_\mu$) in Hopper-med-expert in Figure 1(a). Initially, all twelve ensemble members have the same belief. As the trajectory progresses, the beliefs of each member are updated based on the transition history, with the dominant model (the one with the highest belief) continuously changing. Figure 1(b)-1(d)

further show the belief adaption during imaginary rollouts that start from different time steps of that offline trajectory.

## F.2 COMPARISON WITH MODEL-FREE METHODS ON D4RL MUJOCO

| Data Type | Environment | BA-MCTS (ours) | CQL | BEAR | BRAC-v | SAC | BC |
|---|---|---|---|---|---|---|---|
| random | HalfCheetah | **41.45** (1.40) | 35.4 | 25.1 | 31.2 | 30.5 | 2.1 |
| random | Hopper | **31.42** (0.10) | 10.8 | 11.4 | 12.2 | 11.3 | 1.6 |
| random | Walker2d | **21.54** (0.06) | 7.0 | 7.3 | 1.9 | 4.1 | 9.8 |
| medium | HalfCheetah | **77.31** (1.40) | 44.4 | 41.7 | 46.3 | -4.3 | 36.1 |
| medium | Hopper | **103.9** (0.33) | 86.6 | 52.1 | 31.1 | 0.8 | 29.0 |
| medium | Walker2d | **88.23** (0.78) | 74.5 | 59.1 | 81.1 | 0.9 | 6.6 |
| med-replay | HalfCheetah | **71.24** (2.65) | 46.2 | 38.6 | 47.7 | -2.4 | 38.4 |
| med-replay | Hopper | **106.4** (0.53) | 48.6 | 33.7 | 0.6 | 3.5 | 11.8 |
| med-replay | Walker2d | **91.42** (1.54) | 32.6 | 19.2 | 0.9 | 1.9 | 11.3 |
| med-expert | HalfCheetah | **102.3** (2.27) | 62.4 | 53.4 | 41.9 | 1.8 | 35.8 |
| med-expert | Hopper | 111.8 (0.82) | 111 | 96.3 | 0.8 | 1.6 | **111.9** |
| med-expert | Walker2d | **116.0** (1.49) | 98.7 | 40.1 | 81.6 | -0.1 | 6.4 |
| Average Score | | **80.25** | 54.85 | 39.83 | 31.44 | 4.13 | 25.07 |

Table 9: Comparisons between the proposed algorithms and model-free offline policy learning methods on the D4RL benchmark suite. Each value represents the normalized score, as proposed in Fu et al. (2020), of the policy trained by the corresponding algorithm. These scores are undiscounted returns normalized to approximately range between 0 and 100, where a score of 0 corresponds to a random policy and a score of 100 corresponds to an expert-level policy. For our algorithms, we report the average score of the final ten policy learning epochs and its standard deviation across three different random seeds.

**Here, we compare our algorithms with a series of model-free offline policy learning (Chen et al., 2024) methods in Table 9.** We include SOTA model-free offline RL methods: CQL (Kumar et al., 2020), BEAR Kumar et al. (2019), and BRAC-v (Wu et al., 2019). Additionally, we show the performance of directly applying SAC or Behavioral Cloning (BC) (Chen et al., 2024) to the provided offline dataset in the last two columns. The mean performance of the baselines are taken from related works (Yu et al., 2020; Kidambi et al., 2020; Fu et al., 2020). Our algorithm shows significantly better performance, demonstrating the necessity of model-based learning in these environments.

## F.3 COMPREHENSIVE BENCHMARKING ON D4RL MUJOCO

To firmly establish the state-of-the-art (SOTA) performance of our algorithm, this section provides a comprehensive comparison of BA-MCTS against 19 offline RL methods on the 12 D4RL MuJoCo tasks. To provide context, we briefly categorize the baselines:

- **The model-based (MBRL) methods** include classic approaches like MOPO (Yu et al., 2020), MOReL (Kidambi et al., 2020), and COMBO (Yu et al., 2021), as well as more recent/SOTA methods such as MOBILE (Sun et al., 2023), CBOP (Jeong et al., 2023), RAMBO (Rigter et al., 2022), and MAPLE (Chen et al., 2021b).

- **The model-free (MFRL) methods** cover a wide range of techniques. We include prominent SOTA methods such as the value-based algorithm CQL (Kumar et al., 2020), the implicit Q-learning method IQL (Kostrikov et al., 2021), ensemble-based methods like SAC-N and EDAC (An et al., 2021), the policy constraint method TD3+BC (Fujimoto & Gu, 2021), and the sequence modeling approach Decision Transformer (DT) (Chen et al., 2021a). We also compare against classic MFRL baselines, including BEAR (Kumar et al., 2019), BRAC-v (Wu et al., 2019), and the direct application of SAC and Behavioral Cloning (BC) (Chen et al., 2024). Finally, APE-V (Ghosh et al., 2022) is included as a conceptually related model-free method that also leverages the BAMDP framework.

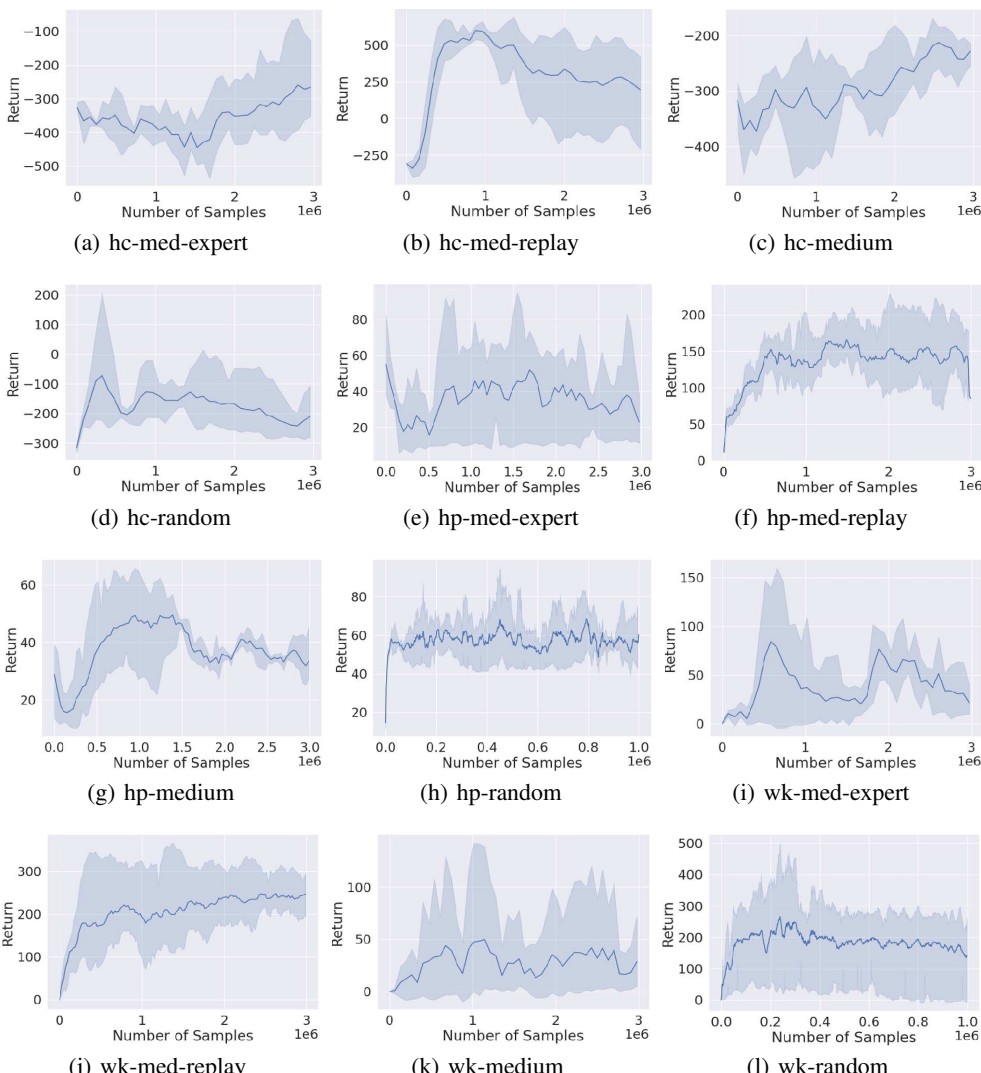

Figure 2: Performance of Sampled EfficientZero on D4RL MuJoCo tasks. The results for HalfChee-tah, Hopper, and Walker2d are presented in the three rows, respectively. Each subfigure depicts the change in undiscounted episodic return as a function of the number of training samples. Experiments are repeated three times with different random seeds, with the solid line representing the mean and the shaded area indicating the 95% confidence interval. For reference, the expert-level episodic returns (corresponding to scores of 100) for HalfCheetah, Hopper, and Walker2d are 12135, 3234.3, and 4592.3, respectively.

The results, including a statistical significance analysis using Cohen's d (Cohen, 1988), are sum-marized in Table 3 in the main content. Note that MOReL, IQL, and the model-free offline RL baselines in Table 9 did not report standard deviations. Additionally, IQL and Decision Transformer (DT) were not evaluated on D4RL tasks with random data.

As shown in Table 3, BA-MCTS achieves the highest average score among all 20 algorithms, both including and excluding the challenging 'random' datasets. To quantify the significance of this im-provement, we compute Cohen's d against all baselines for which standard deviations are available. The results show that the Cohen's d value is greater than 1.8 in all cases, and for most baselines, it is significantly larger (e.g., > 3.5). This indicates that the performance improvement of our algorithm is not only substantial in magnitude but also statistically significant. This comprehensive compari-son provides strong evidence that our proposed method establishes a new state-of-the-art in offline RL on this widely-used benchmark.

| Data Type | Environment | BA-MCTS -SL (ours) | BA-MCTS (ours) | BA-MBRL (ours) | Sampled EfficientZero |
|---|---|---|---|---|---|
| random | HalfCheetah | 11.2 (2.6) | 14.6 (2.2) | 1.2 (0.4) | 54.8 (2.6) |
| random | Hopper | 48.7 (2.4) | 65.2 (1.9) | 5.6 (0.8) | 153.7 (19.4) |
| random | Walker2d | 25.7 (2.0) | 38.1 (1.1) | 5.2 (1.1) | 175.7 (28.2) |
| medium | HalfCheetah | 12.6 (3.0) | 15.3 (1.3) | 1.9 (0.2) | 54.7 (1.9) |
| medium | Hopper | 18.9 (4.8) | 67.5 (0.2) | 1.8 (0.2) | 70.5 (1.0) |
| medium | Walker2d | 24.0 (1.8) | 33.3 (4.0) | 4.8 (1.0) | 63.7 (1.5) |
| med-replay | HalfCheetah | 24.7 (0.9) | 22.4 (1.2) | 2.1 (0.4) | 54.8 (1.8) |
| med-replay | Hopper | 17.9 (0.0) | 12.7 (1.2) | 5.4 (0.2) | 126.6 (12.0) |
| med-replay | Walker2d | 40.3 (7.2) | 77.0 (2.4) | 8.0 (0.3) | 134.9 (4.2) |
| med-expert | HalfCheetah | 32.6 (0.1) | 12.6 (0.8) | 4.7 (1.1) | 55.4 (1.3) |
| med-expert | Hopper | 61.5 (4.6) | 76.9 (11.6) | 5.3 (0.5) | 66.1 (0.9) |
| med-expert | Walker2d | 35.0 (6.5) | 55.9 (1.8) | 2.9 (0.9) | 60.8 (1.8) |

Table 10: Training time (in hours) of our algorithms and Sampled EfficientZero for each evaluation task. Results are presented as the mean and standard deviation from three repeated experiments.

### F.4 COMPARISON WITH MUZERO AND ITS VARIANTS

Monte-Carlo Tree Search (MCTS) (Browne et al., 2012) has been successfully integrated with RL, as exemplified by AlphaZero (Silver et al., 2017) and MuZero (Schrittwieser et al., 2020). These methods have achieved superhuman performance in domains requiring highly sophisticated decision-making processes. AlphaZero relies on a given world model, whereas MuZero learns the world model and policy simultaneously by interacting with the environment. Although there have been various extensions of MuZero (Hubert et al., 2021; Schrittwieser et al., 2021; Ye et al., 2021; Antonoglou et al., 2022; Oren et al., 2022; Zhao et al., 2024), most are designed for online MBRL. **According to Niu et al. (2023), applying MuZero to offline learning, especially for continuous control in highly stochastic environments, remains a significant open challenge.**

Our algorithm design differs from MuZero (and its variants) in several key aspects: (1) MuZero integrates model learning and policy training into a single stage, using a world model defined in a latent state space. Our algorithm separately learns a world model and then trains the policy on top of it, aligning with the widely-adopted offline MBRL framework. (2) MuZero employs a single latent world model (rather than an ensemble) and does not explicitly account for uncertainty in dynamics or reward predictions. This makes it particularly vulnerable to modeling errors in offline MBRL. (3) We introduce double progressive widening (Auger et al., 2013) and Bayes-adaptive planning into MCTS, resulting in a fundamentally different planning algorithm compared to the one used in MuZero.

To evaluate the performance of MuZero-type methods on D4RL MuJoCo tasks, we use the open-source implementation and hyperparameter configurations of Sampled EfficientZero (Ye et al., 2021) provided by LightZero (Niu et al., 2023). Benchmarking results from LightZero indicate that Sampled EfficientZero achieves the best performance on (online) MuJoCo locomotion tasks compared to other MuZero variants. To adapt it for offline learning, we employ the reanalyse technique proposed by Schrittwieser et al. (2021).

The evaluation results are presented in Figure 2. As shown, the performance is significantly worse compared to our methods listed in Figure 3, despite Sampled EfficientZero's higher computational cost. For reference, the expert-level episodic returns (corresponding to scores of 100) for HalfCheetah, Hopper, and Walker2d are 12135, 3234.3, and 4592.3, respectively.

In Table 10, we report the training time of our proposed algorithms and Sampled EfficientZero on the D4RL MuJoCo tasks. The experiments were conducted on a server with 40 Intel(R) Xeon(R) Gold 5215 CPUs and 4 Tesla V100-SXM2-32GB GPUs. Our search-based algorithm, BA-MCTS, requires considerably less training time than Sampled EfficientZero to achieve its superior performance. While BA-MCTS requires more computation than the non-search variant BA-MBRL, this extra cost is limited to the offline training phase; no MCTS is performed during deployment, ensuring real-time execution.

The performance gap may be attributed to the challenges of applying MuZero's design to the offline setting. World model learning is particularly difficult when the state-action space is vast but training data is limited. Sampled EfficientZero integrates model learning and policy training into a single stage, which can significantly increase the learning difficulty compared to our decoupled approach.

### F.5 ABLATION STUDY ON THE ENHANCEMENT FROM BELIEF ADAPTATION AND MCTS-BASED PLANNING

| Data Type | Environment | Optimized | BA-MBRL | BA-MCTS | BA-MCTS-SL |
|-----------|-------------|-----------|---------|---------|------------|
| random | HalfCheetah | 31.7 | 32.76 (1.16) | **36.23** (1.04) | 29.20 (2.00) |
| random | Hopper | 12.1 | 31.47 (0.03) | 31.56 (0.12) | **33.83** (0.10) |
| random | Walker2d | 21.7 | 21.45 (0.53) | 21.59 (0.32) | **21.89** (0.07) |
| medium | HalfCheetah | 45.7 | 56.54 (5.20) | **75.84** (3.81) | 70.47 (3.52) |
| medium | Hopper | 69.3 | **98.25** (3.42) | 96.70 (14.0) | 97.75 (7.09) |
| medium | Walker2d | 79.7 | 75.41 (4.17) | 74.73 (3.25) | **82.84** (1.85) |
| med-replay | HalfCheetah | 58.0 | 62.50 (0.18) | **65.45** (0.81) | 61.16 (1.60) |
| med-replay | Hopper | 90.8 | 93.91 (4.25) | 101.8 (3.46) | **106.3** (0.13) |
| med-replay | Walker2d | 65.8 | **97.54** (1.93) | 95.06 (2.11) | 92.13 (5.13) |
| med-expert | HalfCheetah | **104.2** | 90.52 (4.13) | 76.16 (10.3) | 80.53 (6.63) |
| med-expert | Hopper | 105.8 | 107.8 (0.37) | 108.3 (0.22) | **112.2** (0.29) |
| med-expert | Walker2d | 97.1 | 84.71 (0.87) | **110.0** (1.74) | 107.7 (0.82) |
| Average Score | | 65.2 | 71.06 | 74.45 | **74.62** |

Table 11: Performance comparison among our algorithm: BA-MCTS, its ablated versions: BA-MBRL and BA-MCTS-SL, as well as the baseline: Optimized (Lu et al., 2022). To ensure a principled ablation study, we reimplement BA-MCTS and its ablated variants using the codebase of Optimized, which systematically investigates design choices in offline MBRL. We make only minimal modifications to the code and hyperparameter settings. This approach ensures that any observed performance improvements are attributable to algorithmic design, rather than implementation differences or tuning tricks.

For a principled ablation study, we reimplement BA-MCTS and its ablated variants based on the codebase of Optimized (Lu et al., 2022), which thoroughly explores design choices in offline MBRL, making minimal changes to the code and hyperparameter settings (see Appendix D). The comparison results are presented in Table 11[5], and the training curves of BA-MCTS and its ablated variants are shown in Figure 3.

**Firstly**, by comparing the following two methods, any observed performance difference can be directly attributed to the benefits of belief adaptation in a BAMDP formulation.

- Optimized: A strong, non-adaptive offline model-based RL baseline.
- BA-MBRL: Our ablation variant, which is a minimal modification of Optimized, differing only by the integration of our BAMDP framework that dynamically adapts the belief over the model ensemble via Bayesian updates (Eq. (4)).

As demonstrated, BA-MBRL outperforms the Optimized baseline on 8 out of 12 tasks and achieves a higher average score, providing direct evidence that Bayesian belief adaptation leads to improved final policies.

**Secondly**, by comparing the following two methods, any observed performance difference can be directly attributed to the benefits of incorporating MCTS-based planning:

- BA-MBRL: Our ablation variant that incorporates the BAMDP belief adaptation framework but does *not* use MCTS. Its policy improvement relies solely on Soft Actor Critic.

---

[5]The performance of BA-MCTS decreases, relative to the results presented in Table 2, for two main reasons: (1) The original implementation in Table 2 is based on a different codebase than the one used in Optimized; and (2) for the ablation results shown in Table 11, we adopt the reward penalty design from Optimized to enable exact ablation comparisons, rather than using the one described in Eq. (5).

- BA-MCTS: Our final algorithm, which builds upon BA-MBRL by integrating the Continuous BAMCP deep search planner.

This substantial gain highlights the effectiveness of using Continuous BAMCP as a powerful policy improvement operator. While BA-MBRL relies on the implicit policy improvement provided by the temporal difference learning process, BA-MCTS leverages deep lookahead search to explicitly find a stronger, non-parametric policy ($\pi_{\text{ret}}$ returned by Algorithm 1) at each decision point. The actor-critic learner is then trained on the high-quality trajectories generated from this improved policy. This "RL+Search" paradigm, where the learner distills knowledge from a stronger planner, is the primary reason for the observed performance leap. This result strongly supports our claim that the deep search component is crucial for achieving state-of-the-art performance in the offline model-based setting.

### F.6 Comparison of Policy Distillation Mechanisms

This section addresses the research question: How can the search outcomes (i.e., $\pi_{\text{ret}}$) be effectively used for policy improvement? We investigate this by comparing two different mechanisms for distilling the knowledge from the MCTS planner back into the parameterized policy network $\pi$.

- BA-MCTS: Our primary algorithm, which uses a policy gradient method (i.e., SAC) to update the policy network using trajectories generated by the planner.
- BA-MCTS-SL: An alternative variant where the policy network is updated via supervised learning (SL) to directly mimic the planner's output distribution $\pi_{\text{ret}}$ by minimizing a cross-entropy loss.

**Performance Comparison:** As shown in Table 11, BA-MCTS-SL performs similarly to BA-MCTS in terms of the final average score, validating the effectiveness of both policy update mechanisms. This indicates that both policy gradient and supervised learning are viable methods for distilling the results of the deep search. However, we note that BA-MCTS-SL can be less stable on certain complex tasks; for example, it struggles on the Walker2d environments, where a warm-up training phase (using BA-MBRL) is required to establish a good initial policy for the supervised learning to be effective.

**Training Stability and Robustness:** On the other hand, a key advantage of the SL-based policy update is its training stability. As evident in the training plots (Figure 3), BA-MCTS-SL often exhibits much smoother learning curves compared to the other two algorithms, indicating greater robustness in model selection for deployment. Furthermore, as shown in our ablation study on the reward penalty (Appendix F.7), we found that the SL-based policy update is less sensitive to the degree of pessimism.

### F.7 Ablation Study on the Reward Penalty

| Data Type | Environment | BA-MCTS -SL | BA-MCTS | BA-MBRL | BA-MCTS -SL ($\lambda = 0$) | BA-MCTS ($\lambda = 0$) | BA-MBRL ($\lambda = 0$) |
|---|---|---|---|---|---|---|---|
| random | HalfCheetah | 29.20 (2.00) | 36.23 (1.04) | 32.76 (1.16) | 34.80 (1.39) | 38.78 (1.65) | **39.64** (2.86) |
| random | Hopper | **33.83** (0.10) | 31.56 (0.12) | 31.47 (0.03) | 9.16 (0.16) | 7.44 (0.14) | 6.97 (0.07) |
| random | Walker2d | **21.89** (0.07) | 21.59 (0.32) | 21.45 (0.53) | 17.53 (6.16) | 21.53 (0.42) | 21.41 (0.64) |
| medium | HalfCheetah | 70.47 (3.52) | **75.84** (3.81) | 56.54 (5.20) | 61.64 (4.58) | 60.84 (2.00) | 41.49 (2.29) |
| medium | Hopper | 97.75 (7.09) | 96.70 (14.0) | 98.25 (3.42) | 102.8 (2.29) | **104.4** (1.88) | 93.68 (11.4) |
| medium | Walker2d | 82.24 (1.85) | 74.73 (3.25) | 75.41 (4.17) | **82.61** (0.86) | 57.01 (7.24) | 57.97 (15.4) |
| med-replay | HalfCheetah | 61.16 (1.60) | **65.45** (0.81) | 62.50 (0.18) | 42.10 (2.85) | 36.65 (2.39) | 44.03 (7.35) |
| med-replay | Hopper | 106.3 (0.13) | 101.8 (3.46) | 93.91 (4.25) | **107.9** (0.07) | 84.11 (2.97) | 91.81 (11.5) |
| med-replay | Walker2d | 92.13 (5.13) | 95.06 (2.11) | 97.54 (1.93) | 88.61 (5.21) | 97.33 (3.51) | **98.19** (1.23) |
| med-expert | HalfCheetah | 80.53 (6.63) | 76.16 (10.3) | **90.52** (4.13) | 51.76 (5.31) | 26.60 (1.46) | 29.88 (2.28) |
| med-expert | Hopper | **112.2** (0.29) | 108.3 (0.22) | 107.8 (0.37) | 106.8 (6.34) | 81.76 (6.45) | 86.79 (18.7) |
| med-expert | Walker2d | 107.7 (0.82) | 110.0 (1.74) | 84.71 (0.87) | **110.8** (1.72) | 110.2 (0.91) | 53.35 (38.0) |
| Average Score | | **74.62** | 74.45 | 71.06 | 68.04 | 60.55 | 55.43 |

Table 12: Comparison of the proposed algorithms with their corresponding versions without the reward penalty (i.e., $\lambda = 0$). The definitions of the scores in this table are consistent with those in Table 9.

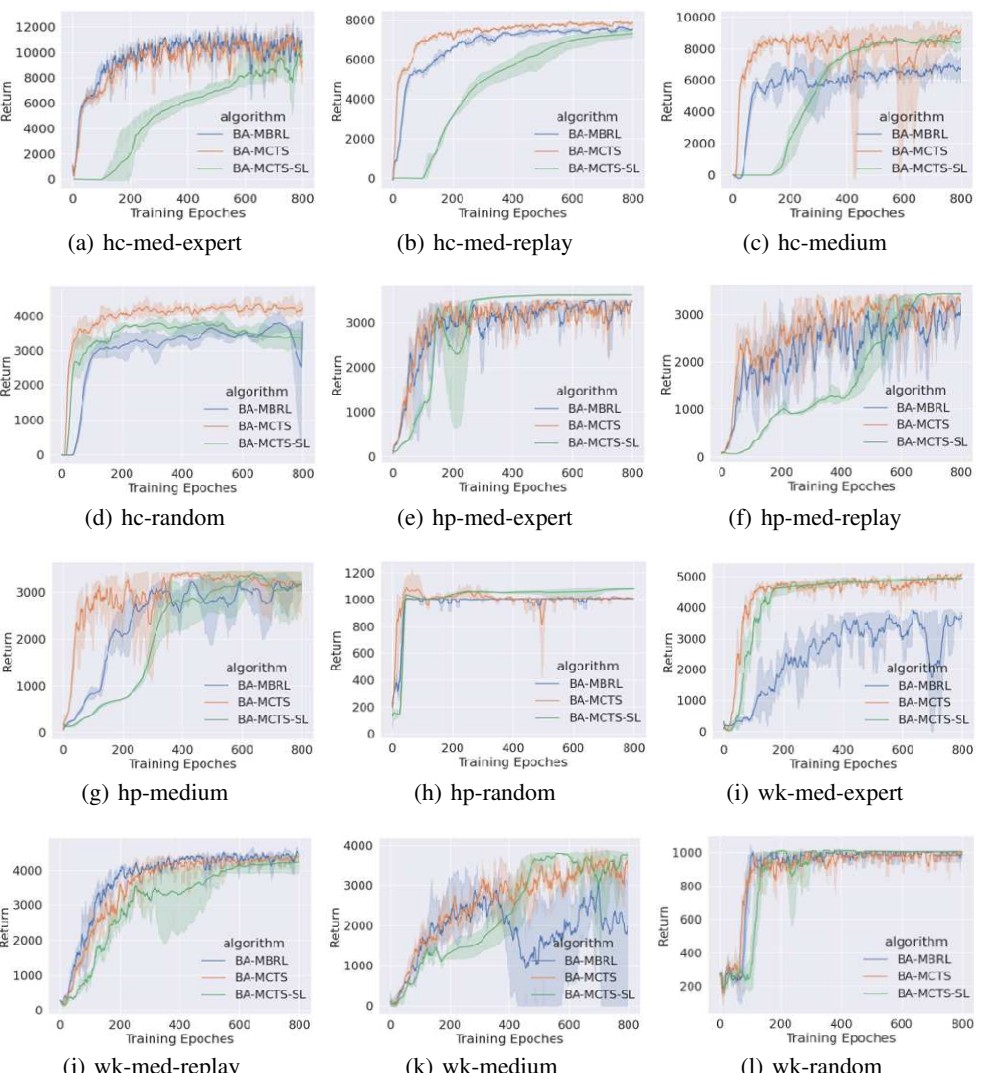

Figure 3: Performance of our proposed algorithms on D4RL MuJoCo tasks in the ablation study. The results for HalfCheetah, Hopper, and Walker2d are presented in the three rows, respectively. Each subfigure depicts the change in the undiscounted episodic return as a function of training epochs. Experiments are repeated three times with different random seeds, with the solid line representing the mean and the shaded area indicating the 95% confidence interval. For reference, the expert-level episodic returns for HalfCheetah, Hopper, and Walker2d are 12135, 3234.3, and 4592.3, respectively. Note that the training epochs for each algorithm, as listed in Table 5, have been linearly scaled to 800 for better visualization.

To demonstrate the necessity of incorporating the reward penalty in offline MBRL, we conduct an ablation study by setting $\lambda$ in Eq. (5) to 0, resulting in ablated versions of our proposed three algorithms. The results are presented in Table 12. **First**, the average performance of the algorithms with the reward penalty is consistently better, demonstrating the importance of using reward penalties in offline MBRL to prevent the overexploitation of the learned world models (which can be inaccurate). **Second**, the supervised-learning-based algorithm (i.e., BA-MCTS-SL ($\lambda = 0$)) is less affected by the absence of the reward penalty, compared to the policy-gradient-based methods. Notably, BA-MCTS-SL ($\lambda = 0$) and BA-MCTS-SL achieve comparable performance in the Hopper and Walker2d tasks. This shows an additional advantage of BA-MCTS-SL – its reduced sensitivity to model inaccuracies. **Lastly**, there are instances where superior performance is achieved with $\lambda$ set to 0. For example, BA-MCTS-SL ($\lambda = 0$) performs better than BA-MCTS-SL in 5 out of 12

tasks. This suggests that the performance of our algorithms in Tables 11 could be further improved by adjusting hyperparameters such as $\lambda$.

## G  STATEMENT ON THE USE OF LARGE LANGUAGE MODELS

In the spirit of transparency and in accordance with the ICLR disclosure policy, we detail the use of Large Language Models (LLMs) in the preparation of this manuscript.

LLMs were utilized as an assistive tool in two main capacities: language refinement and technical support during the formalization of proofs. The specific roles were:

- **Language and Readability Enhancement**: We employed an LLM to polish the manuscript's language. This included improving grammar, rephrasing sentences for better clarity, and ensuring a consistent and professional tone throughout the paper.

- **Assistance with Theoretical Proofs**: An LLM provided support in the technical writing and verification of our theoretical proofs. This assistance was twofold:
    1. Generating LaTeX source code for complex mathematical equations.
    2. Acting as a verification aid by reviewing the logical steps in our derivations to help identify potential inconsistencies or errors.

It is crucial to state that the LLM did not contribute to the original ideation, conceptual development, or the core derivations of the proofs themselves. The intellectual ownership and the scientific reasoning behind all theoretical results remain entirely with the authors. The authors have meticulously reviewed, validated, and take full and final responsibility for all content, including the correctness and rigor of the proofs that were checked with LLM assistance.

