# OpenReview forum: "Bayes Adaptive Monte Carlo Tree Search for Offline Model-based Reinforcement Learning"
_ICLR.cc/2026/Conference — ICLR 2026 Poster_

### Official Review · Reviewer_JBcC · 2025-10-30

**Soundness:** 2
**Presentation:** 2
**Contribution:** 2
**Rating:** 6
**Confidence:** 4

**Summary:**

This paper frames offline MBRL as a BAMDP to address model uncertainty inherent in learning from offline datasets. The authors propose a novel Continuous BAMCP planning algorithm based on Monte Carlo Tree Search with double progressive widening for continuous domains. Integrating this with a policy iteration scheme, they present a framework termed BA-MCTS that distills the search/planning output into policy/value networks for efficient real-time decision making. Empirical evaluations are provided on D4RL benchmark, showing strong performance relative to several baselines.

**Strengths:**

1.Integrating BAMCP with offline MBRL and extending it to continuous spaces via the introduction of DPW is well-motivated and conceptually sound.
2.The paper includes well-executed ablation studies that systematically evaluate the individual contributions of the algorithm’s components, highlighting the advantages of the BAMDP formulation, MCTS planning, and the search-guided policy learning approach.
3.Theorem 4.1 establishes a consistency guarantee for the planning algorithm, proving its convergence to the optimal value in the pseudo-pessimistic BAMDP framework.

**Weaknesses:**

1.Unclear and inconsistent notation for augmented states: The notation used to represent augmented states—such as $(s, h)$ versus $(s, b)$—lacks clarity, and the distinction between history, belief, and memory storage in tree nodes remains vague. This ambiguity, evident across equations, algorithm steps (e.g., Algorithm 1, Page 5), and explanatory text, may cause confusion, particularly for readers less familiar with BAMCP/POMCP.
2.Insufficient contextualization within Bayesian and offline MBRL literature: The paper would benefit from a more thorough discussion connecting the proposed method to existing frameworks that model offline RL as a Bayes Adaptive MDP or epistemic POMDP, such as those presented in Ghosh et al. (2022) and Dorfman et al. (2021).
Ghosh et al. (2022): "Offline RL policies should be trained to be adaptive."
Dorfman et al. (2021): "Offline meta reinforcement learning – identifiability challenges and effective data collection strategies."

**Questions:**

1.In Table 1, should the mean prediction errors be calculated as weighted averages based on the dimensions of states and rewards? Why does the Walker2d environment exhibit such large prediction errors in Table 1, while showing very small performance variance in Table 2?
2.Why are the results reported in Table 2 inconsistent with those in Table 12?

---

> ### Author Response · Authors · 2025-11-20
>
> ## Regarding Weakness 1:
>
> We sincerely thank the reviewer for pointing out this critical inconsistency in our notation. We agree that the interchangeable use of $(s, b)$ and $(s, h)$ can be confusing and we apologize for obscuring the implementation details.
>
> To clarify, $(s, b)$ represents the information state in the theoretical BAMDP formulation (Section 2 & 4.1), where $s$ is the physical state and $b$ is the belief distribution over model parameters $\theta$. In contrast, $(s, h)$ is used in the MCTS planning context (Section 4.2) to uniquely identify a node in the search tree. Since every node in the tree is reached via a unique trajectory from the root, the history $h$ naturally distinguishes the specific decision point in the search process.
>
> Crucially, the belief $b$ is derived directly from the history $h$. As mentioned in Footnote 3 (Page 6), $b$ is a function of $h$, recursively updated using the transition data contained in $h$ via the Bayesian posterior update rule (Eq. 4). In our actual implementation, while the node is conceptually indexed by its history $(s,h)$ to define its position in the tree, the node data structure explicitly stores the physical state $s$, the current belief distribution $b$, and the list of valid actions. The belief $b$ serves as the sufficient statistic of the history $h$ for making predictions.
>
> In the final version, we will rigorously unify these notations and add a specific remark in Section 4 to explicitly clarify the relationship between history $h$ and belief $b$, as well as how they are stored within the tree nodes, to ensure clarity for readers less familiar with BAMCP.
>
> ## Regarding Weakness 2:
>
> We thank the reviewer for this valuable feedback.
>
> Regarding the connection to Ghosh et al. (2022), we would like to clarify that we have cited this work and compared our method against it in the submitted manuscript. In our paper, their method is referred to as APE-V, and the comparison is detailed in Section 5.2 and reported in Tables 2 and 3.
>
> We will add a paragraph to the related works of our paper, to incorporate research on Bayesian approaches to offline RL and algorithms leveraging model-based search outcomes to enhance the efficiency of actor-critic training. We agree that including these prior works adds clarity to our contributions.
>
> > **Proposed Addition to Related Works:**
> >
> > To sum up, our algorithm introduces a Bayesian approach to offline MBRL and leverages tree search to enhance policy learning. There has been related research in both directions. (1) Dorfman et al. (2021); Choshen & Tamar (2023) propose to model offline Meta RL as a BAMDP and learn a belief-conditioned policy capable of adapting to different underlying MDPs for multi-task purposes. Ghosh et al. (2022) apply the BAMDP framework to model-free offline RL, arguing that optimal policies for offline RL should be adaptive to all observed transitions. Nevertheless, these works do not explore the Bayesian treatment of model-based RL. (2) Model-based planning results can be utilized to improve the sample efficiency of model-free RL. For instance, Feinberg et al. (2018) propose Model-based Value Expansion. It uses the learned world model to generate imaginary rollouts, providing a more accurate estimation of value function targets for online actor-critic training. This idea is later extended to the offline RL setting by Jeong et al. (2023). However, they do not employ BAMDP for uncertainty treatment, and, compared to model-based rollouts, MCTS can offer more exhaustive exploration, crucial for tackling complex tasks.
>
> ## Reference
> Era Choshen and Aviv Tamar. Contrabar: Contrastive bayes-adaptive deep RL. In *International Conference on Machine Learning*, volume 202, pp. 6005–6027. PMLR, 2023.
>
> Ron Dorfman, Idan Shenfeld, and Aviv Tamar. Offline meta reinforcement learning - identifiability challenges and effective data collection strategies. In *Advances in Neural Information Processing Systems*, pp. 4607–4618, 2021.
>
> Vladimir Feinberg, AlvinWan, Ion Stoica, Michael I Jordan, Joseph E Gonzalez, and Sergey Levine. Model-based value expansion for efficient model-free reinforcement learning. In *International Conference on Machine Learning*, 2018.
>
> Dibya Ghosh, Anurag Ajay, Pulkit Agrawal, and Sergey Levine. Offline RL policies should be trained to be adaptive. In *International Conference on Machine Learning*, volume 162, pp. 7513–7530. PMLR, 2022.
>
> Jihwan Jeong, XiaoyuWang, Michael Gimelfarb, Hyunwoo Kim, Baher Abdulhai, and Scott Sanner. Conservative bayesian model-based value expansion for offline policy optimization. In *International Conference on Learning Representations*. OpenReview.net, 2023.

---

> ### Author Response · Authors · 2025-11-20
>
> ## Regarding Question 1
>
> Regarding the calculation method in Table 1, we clarify that the prediction error is defined as the mean L2 Norm of the difference between the ground truth and predicted state vectors within the rollouts, rather than a weighted average across dimensions.
>
> Regarding the observation in the Walker2d environment, we believe that the stability of policy learning in this task does not strictly require highly accurate state predictions. Despite the relatively high prediction error for states, our algorithm successfully converges to a robust policy, resulting in the small performance variance observed in Table 2.
>
> Finally, we assure you that the results in Table 1 and Table 2 are authentic and fully reproducible. The reproduction code for both tables, as well as the raw data for Table 1, are provided in the submitted supplementary material.
>
> ## Regarding Question 2:
>
> We thank the reviewer for their sharp observation and apologize for the confusion caused by this discrepancy.
>
> The difference in results is intentional and has been currently noted in Footnote 5 (page 30).
>
> - Table 2 reports the performance of our final, state-of-the-art algorithm configuration. This version uses the "Optimized" (Lu et al., 2022) codebase and the specific uncertainty-based reward penalty described in Eq. (5).
>
> - Table 12 (and Table 11), on the other hand, presents results from a strictly controlled ablation study. To ensure a principled
> ablation study, we reimplement BA-MCTS and its ablated variants using the codebase of Optimized, which systematically investigates design choices in offline MBRL. This approach ensures that any observed performance improvements are attributable to algorithmic design, rather than implementation differences or tuning tricks.
>
> Therefore, the "BA-MCTS" reported in Table 12 (Average Score 74.45) is effectively a constrained version used solely for internal comparison, whereas Table 2 (Average Score 80.25) represents the full potential of our proposed method.
>
> In the revision, we will move this explanation from the footnote to the main text of Section 5.3 and the captions of Tables 11 and 12 to ensure this distinction is immediately clear to readers.
>
> ## Reference
> Cong Lu, Philip J. Ball, Jack Parker-Holder, Michael A. Osborne, and Stephen J. Roberts. Revisiting design choices in offline model based reinforcement learning. In *International Conference on Learning Representations*. OpenReview.net, 2022.

---

> > ### Comment · Reviewer_JBcC · 2025-11-27
> >
> > I thank the authors for their responses and for the revisions made to the manuscript. They have adequately addressed the main issues I raised. I have no further significant concerns.

---

### Official Review · Reviewer_u5So · 2025-10-31

**Soundness:** 3
**Presentation:** 2
**Contribution:** 2
**Rating:** 4
**Confidence:** 3

**Summary:**

This paper views offline RL as a Bayes Adaptive Markov Decision Process (BAMDP) for effectively addressing uncertainty associated with a finite dataset, and an associated planning algorithm based on MCTS that is then folded back into learning a policy using a policy search procedure. Empirical simulations show that the proposed algorithm outperforms baselines in offline RL benchmark suites.

**Strengths:**

- Using search within offline RL based policy search is pretty interesting as a general approach, and this paper makes progress towards marrying these ideas in a clean manner. That said, my knowledge of this area isn't up to date particularly with the more recent efforts and so I do not know if there are similar efforts in offline RL that I am unaware of.

**Weaknesses:**

- The writing at times appears to come in thick and fast, and in a sense i understand why this happens given the different sub-routines but it does help to work a bit more to help clarity in exposition.

**Questions:**

- One thing that wasn't clear was is the output of the learner just the policy?
- What do the tradeoffs look like if we tried to use the search procedure even at inference time, somewhat mirroring MPC style approaches?
- For me, it wasnt totally clear what was the net positive from trying to use search in this setup, as it does appear as if one can effectively solve the problem using a mix of pessimism and dyna type sampling, and perhaps this isnt totally clear just from the current exposition.

---

> ### Author Response · Authors · 2025-11-20
>
> We thank the reviewer for the valuable feedback. We appreciate that you recognized the comprehensive nature of our sub-routines. Below, we outline our plan to improve the clarity of exposition and address your specific questions regarding the system outputs and the search component.
>
> ## Regarding Weakness: Clarity of Exposition
>
> We acknowledge that the paper involves multiple dense components. To address this in the revision, we will add a schematic overview diagram in Section 4 to visually illustrate the data flow: **Offline Data $\to$ Ensemble Learning $\to$ Construct BAMDP $\to$ MCTS Planning (Data Generation) $\to$ Policy Distillation**. This will help readers better navigate the connection between the sub-routines (Continuous BAMCP) and the overall framework (BA-MCTS).
>
> ## Regarding Question 1:
>
> Yes, the final output of the learner is solely the policy network.
>
> While our framework employs a model ensemble and MCTS planning during the offline training phase to generate high-quality data for policy improvement, these components are not used during deployment. The knowledge from the search is distilled into the policy network. This design ensures that the inference process is lightweight and capable of real-time execution (e.g., for high-frequency Tokamak control), avoiding the computational overhead of running search or ensemble updates online.
>
> ## Regarding Question 2:
>
> Using the search procedure at inference time (similar to MPC) presents significant trade-offs regarding latency and deployment complexity. Our system is targeted at tasks like nuclear fusion control, which require high control frequencies (25-50Hz). Running Continuous BAMCP online is computationally intensive and would drastically reduce the control frequency, making it infeasible for such real-time applications. Furthermore, it increases deployment complexity: an online search approach requires deploying the entire ensemble of world models and the planner to the control hardware, whereas our approach only requires the lightweight policy network.
>
> ## Regarding Question 3:
>
> We address the benefit of search from three perspectives:
> 1. **Theoretical Motivation:** Our motivation is grounded in the problem formulation itself. In Section 4.1, we explicitly model the model uncertainty in Offline MBRL as a Bayes-Adaptive MDP (BAMDP). As established by prior work (e.g., BAMCP [1] ), MCTS is a theoretically grounded and efficient method for solving BAMDPs. Thus, using search is a natural consequence of our rigorous formulation.
> 2. **Superiority over Dyna-style Baselines:** Empirically, Table 2 shows that our algorithm consistently outperforms a wide range of state-of-the-art model-based methods. Notably, strong baselines like MOBILE [2] , RAMBO [3] , MAPLE [4] , and Optimized [5] all employ the "pessimism + Dyna-type sampling" approach mentioned by the reviewer. However, their performance is consistently inferior to our method, suggesting that our approach provides significant advantages over standard Dyna-based methods.
> 3. **Direct Ablation Evidence:** Most critically, we validated this in our ablation study (Section 5.3 and Appendix F.5) by comparing BA-MCTS against its variant BA-MBRL (which removes the search component while retaining the Bayesian belief adaptation). As shown in Table 11, BA-MCTS achieves a substantial performance gain over BA-MBRL. This highlights the effectiveness of Continuous BAMCP as a powerful policy improvement operator: while BA-MBRL relies on implicit improvement via TD learning, BA-MCTS leverages deep lookahead search to explicitly find a stronger policy ($\pi\_{ret}$), providing higher-quality targets for the learner. This "RL+Search" paradigm is the primary reason for the observed performance leap.
>
> ## Reference
> [1] Arthur Guez, David Silver, and Peter Dayan. Scalable and efficient bayes-adaptive reinforcement learning based on monte-carlo tree search. Journal of Artificial Intelligence Research, 48:841– 883, 2013.
>
> [2] Yihao Sun, Jiaji Zhang, Chengxing Jia, Haoxin Lin, Junyin Ye, and Yang Yu. Model-bellman inconsistency for model-based offline reinforcement learning. In International Conference on Machine Learning, volume 202, pp. 33177–33194. PMLR, 2023.
>
> [3] Marc Rigter, Bruno Lacerda, and Nick Hawes. RAMBO-RL: robust adversarial model-based offline reinforcement learning. In Advances in Neural Information Processing Systems, 2022.
>
> [4] Xiong-Hui Chen, Yang Yu, Qingyang Li, Fan-Ming Luo, Zhiwei (Tony) Qin, Wenjie Shang, and Jieping Ye. Offline model-based adaptable policy learning. In Advances in Neural Information Processing Systems, pp. 8432–8443, 2021b.
>
> [5] Cong Lu, Philip J. Ball, Jack Parker-Holder, Michael A. Osborne, and Stephen J. Roberts. Revisiting design choices in offline model based reinforcement learning. In International Conference on Learning Representations. OpenReview.net, 2022.

---

### Official Review · Reviewer_Gn3V · 2025-11-01

**Soundness:** 4
**Presentation:** 4
**Contribution:** 4
**Rating:** 8
**Confidence:** 4

**Summary:**

This paper reframes Offline Model-Based Reinforcement Learning (MBRL) as a Bayes-Adaptive Markov Decision Process (BAMDP) to formally handle uncertainty over learned dynamics models.
It introduces Continuous BAMCP, a Monte-Carlo Tree Search planner extended to continuous, stochastic domains using Double Progressive Widening (DPW), and integrates it into a policy-iteration framework named BA-MCTS.  Maintain a belief distribution over ensemble models and update it via Bayesian posterior rules.  Provide a *consistency theorem* guaranteeing exponential convergence of the planner’s value estimate. Empirically achieve convincing results on D4RL MuJoCo tasks and 3 Tokamak control tasks.

**Strengths:**

1. Novel idea of combining BAMDP formulation, MCTS planning, and offline MBRL into a unified algorithmic pipeline.
2. Provides a **consistency proof** for Bayes-adaptive planning in continuous spaces.
3. Bayesian belief updates give interpretable, adaptive weighting over model ensembles
4. Reward-variance penalty connects naturally to pessimistic (safe) RL literature
5. Strong empirical performance
6. Generalization: Demonstrates transfer to real-world-style stochastic control (Tokamak tasks).

**Weaknesses:**

1. **Computation cost:** Continuous BAMCP remains expensive, many simulations per state, not scalable to large or long-horizon domains.
2. **Limited data efficiency analysis:** No study of how ensemble size or dataset quality affect performance.
3. **Offline-only limitation:** Once trained, policy does not update beliefs from online experience.

**Questions:**

- What is the computational cost (simulations per decision, wall-time) relative to other baselines?
- Could Continuous BAMCP be applied in learned latent spaces to improve scalability?

---

> ### Author Response · Authors · 2025-11-20
>
> We sincerely thank you for your positive assessment and constructive comments. We are encouraged that you found our paper well-organized and recognized the value of our contributions. Below, we address your specific concerns.
>
> ## Regarding Weakness 1 & Question 1: Computational Cost and Scalability
>
> **Deployment Cost:** We clarify that the computationally intensive Continuous BAMCP is employed strictly during the offline training phase to generate high-quality targets for policy learning. During deployment, we only execute the learned policy network, ensuring real-time performance with no additional computational overhead compared to standard RL policies.
>
> **Training Cost:** As detailed in Appendix F.4 and Table 10, BA-MCTS requires significantly less training time than Sampled EfficientZero [1] (a SOTA search-based baseline) while achieving superior performance. Although BA-MCTS incurs higher training costs than non-search baselines (e.g., BA-MBRL), we believe this one-time offline cost is justified by the substantial performance gains.
>
> **Scalability:** Regarding scalability, our algorithm is specifically designed for continuous state and action spaces. Unlike standard discrete MCTS, Continuous BAMCP utilizes Double Progressive Widening to effectively handle infinite branching factors, making it inherently scalable to high-dimensional continuous control tasks.
>
> ## Regarding Weakness 2:
>
> - Regarding data quality, we extensively evaluated our method across four distinct dataset quality levels (random, medium, medium-replay, and medium-expert) within the D4RL benchmark.
> - Regarding ensemble size, as noted in Section 5.3, we adopted the exact hyperparameter configuration (including ensemble size) from the "Optimized" [2] baseline to ensure a fair comparison. Since the impact of ensemble size on offline MBRL performance was thoroughly investigated in that work, we did not repeat the analysis here.
>
> ## Regarding Weakness 3:
>
> We clarify that the belief $b(\theta)$ represents the distribution over the ensemble of dynamics models used for planning. During the online deployment phase, the agent executes the learned policy network directly and does not utilize the model ensemble. Therefore, updating the belief based on online experience is not required for the policy's execution.
>
> ## Regarding Question 2:
>
> This is an excellent suggestion. We agree that planning in a learned latent space is a promising direction for improving scalability, similar to MuZero [3]. In this work, we prioritized the original state space to ensure direct and interpretable Bayesian belief updates (Eq. 4) on the ensemble, and to avoid compounding errors from latent representation learning. However, integrating our Bayes-Adaptive framework with latent space planning is a compelling avenue for future research to combine scalability with principled uncertainty management.
>
> ## Reference
> [1] Weirui Ye, Shaohuai Liu, Thanard Kurutach, Pieter Abbeel, and Yang Gao. Mastering atari games with limited data. In Advances in Neural Information Processing Systems, pp. 25476–25488, 2021.
>
> [2] Cong Lu, Philip J. Ball, Jack Parker-Holder, Michael A. Osborne, and Stephen J. Roberts. Revisiting design choices in offline model based reinforcement learning. In International Conference on Learning Representations. OpenReview.net, 2022.
>
> [3] Julian Schrittwieser, Ioannis Antonoglou, Thomas Hubert, Karen Simonyan, Laurent Sifre, Simon Schmitt, Arthur Guez, Edward Lockhart, Demis Hassabis, Thore Graepel, Timothy P. Lillicrap, and David Silver. Mastering atari, go, chess and shogi by planning with a learned model. Nature, 588(7839):604–609, 2020.

---

### Official Review · Reviewer_jrky · 2025-11-03

**Soundness:** 3
**Presentation:** 4
**Contribution:** 3
**Rating:** 6
**Confidence:** 3

**Summary:**

This paper introduces a novel framework for offline model-based reinforcement learning, which (1) trains an ensemble of models based on data from a behavior policy and builds a Bayes Adaptive MDP (BAMDP) of this ensemble (i.e., an MDP with uncertainty, where the belief about the transition and reward distributions is updated based on observed transitions); (2) introduces a shaped reward for the Bayes Adaptive MDP that penalizes actions with high uncertainty, called Pessimistic Bayes-Adaptive MDP; (3) extends an existing online-planning algorithm for BAMDP, called BAMCP, to handle continuous states and actions based on Double Progressive Widening; (4) proves that this extension, called Continuous BAMCP, approximates the optimal policy; and finally, (5) introduces an algorithm for efficiently learning a policy from the planning outcomes of Continuous BAMCP.

**Strengths:**

* The paper introduces a number of novel ideas for offline model-based RL: formulating an ensemble of learned models as a Bayes Adaptive MDP, extending BAMDP to handle continuous state and action spaces, and distilling the planning outcomes into a policy. According to the numerical results, most of these ideas are helpful in practice.
* The proposed extension of BAMDP, called Continuous BAMDP, is proven to converge.
* The proposed framework is compared to offline RL baselines on benchmark environments and data, and the numerical results are promising. There is also an ablation study.
* The paper is generally well organized and easy to follow.
* The source code is available, and it supported by some documentation.

**Weaknesses:**

* My main concern is that treating the ensemble of learned models as a Bayes Adaptive MDP for offline RL does not seem to make sense conceptually. Bayes Adaptive MDP is perfectly reasonable when the beliefs are updated based on transitions that are sampled from the real environment; each transition can provide new information, which we can use to update our beliefs (i.e., probabilities or probability densities associated to particular MDPs). However, in the proposed framework, the transitions are sampled from the learned models. What new information do these transitions provide? If we set our prior belief to be a uniform distribution, and then uniformly sample transitions according to these models, why is our expected updated belief different from uniform? For a particular sampled transition, the updated belief will of course be non-uniform; but why would it be non-uniform in expectation? And if it is, then are the belief updates just driven by randomness (e.g., probability of the model that was sampled uniformly at random would likely be increased)? How is this belief update useful?
* That being said, Table 1 in the main text does show that the Bayesian update significantly improves predictions for imaginary rollouts (compared to ground-truth rollouts in the real environments for the same actions sequences). So, for some reason, the Bayesian update is useful; but it is not obvious why. What is the explanation? Does the improvement hinge on the particular ensemble of neural-network models? Does it hinge on the characteristics of these neural networks? Does it give an advantage to neural networks whose output distributions are more concentrated? If yes, could the Bayesian approach be replaced with simply favoring such models? Or does the Bayesian update help by simply concentrating the beliefs to a very small support after a few steps (as shown by the numerical results), which is somehow better for the simulation? If yes, could the Bayesian approach be replaced by simply using one particular randomly chosen model for each episode?
* In the end, the numerical results do support the proposed approach. But it would be good to clarify why it works. (Appendix F.1 is interesting, but does not address the core question.)
* The paper includes a number of ideas, addressing various challenges. While the paper is overall well organized and easy to read, it might benefit from a clearer focus. For instance, the proposed framework would be novel and complete even without Continuous BAMCP; it would simply be restricted to discrete environments. The policy learning also seems somewhat optional since the framework could be presented as an online planning approach (based on model learned from offline experiences); according to the numerical results (Table 11 in the appendix), BA-MCTS and BA-MCTS-SL are pretty close, so the policy learning does not seem that important compared to the other parts.

**Questions:**

* Can you please explain why the Bayesian update works in the offline setting?

---

> ### Author Response · Authors · 2025-11-20
>
> ## Regarding Weakness 1 & 3 and Question 1: Conceptual Validity of Offline Bayesian Updates
>
> We thank the reviewer for this thoughtful question. We clarify that the belief update in our framework consists of two distinct phases, each serving a different purpose:
>
> **Phase 1: Belief Calibration via Real History.**
>
> This phase occurs before the ACTOR (Algorithm 2) initiates a new imaginary rollout. When we sample a starting state $s$ from the offline dataset $\mathcal{D}\_\mu$, we retrieve its corresponding real trajectory history $h\_{real} = (s_0, a_0, \dots, s_t=s)$. We use this real history to perform the Bayesian update (Eq. 4) starting from a uniform prior $b_0$, resulting in a calibrated "starting belief" $b_{start} = b(\theta | h_{real})$. This update is valid because it utilizes real-world information. The process effectively "calibrates" our ensemble, identifying which models perform better on the actual observed history. This $b_{start}$ (rather than a uniform distribution) is then passed as the prior to the SEARCH procedure.
>
> **Phase 2: Temporal Consistency during Planning.**
>
> This phase occurs during the imaginary rollout updates. We acknowledge that this update does not provide new information about the real world. Instead, its purpose is different: it acts as a planning heuristic to maintain temporal consistency within a single MCTS simulation. If we did not update $b_{imag}$, MCTS would sample from the mixture of all models $ \sum b\_{start}^i \mathcal{P}\_{\theta}^i$ at every step. This would be akin to "re-rolling the dice" to decide the world model at every single time step. By dynamically updating $b_{imag}$, the simulated trajectory $(s'\_{imag}, s''\_{imag}, \dots)$ tends to align with a single, internally consistent MDP dynamic. Furthermore, as noted in the paper (Section 4.3), the segment length $H$ is kept short precisely to minimize error accumulation when interacting with learned models.
>
> **Summary:** Phase 1 utilizes real-world data to ensure reliability. Phase 2 ensures temporal consistency during the selection of ensemble members. Together, they guarantee the effectiveness of our belief update mechanism.
>
> ## Regarding Weakness 2
>
> **On the question: "Does it give an advantage to neural networks whose output distributions are more concentrated?"**
>
> It is not necessarily true that the Bayesian update favors models with more concentrated output distributions. A concentrated distribution does not imply correctness; a model can be confident but inaccurate. Our algorithm updates beliefs based on the likelihood of the observed data. Therefore, it favors models that accurately explain the transition history, regardless of whether their output distributions are concentrated or dispersed.
>
> **On the question: "Could the Bayesian approach be replaced by simply using one particular randomly chosen model for each episode?"**
>
> We argue that there is not a principled manner to choose such a specific model for an entire episode. A randomly selected model (even if sampled from the prior) might perform poorly on the specific trajectory history leading to the current state. Relying on such a model would produce unreliable state transitions during planning.
>
> **Summary:** We employ Bayesian Planning not as an arbitrary choice, but because we have formally defined the model uncertainty problem in Offline MBRL as a Bayes-Adaptive MDP (Section 4.1). This framework necessitates a principled belief update mechanism rather than heuristic model selection.
>
> ## Regarding Weakness 4: Motivation and Necessity of Components
>
> **Necessity of Continuous BAMCP** Our primary motivation is solving data-driven control problems, specifically Tokamak plasma control. In such domains, no simulator is available, necessitating Offline MBRL. We introduce the BAMDP framework to rigorously model the model uncertainty inherent in this setting. Crucially, these are continuous control problems, meaning we are tasked with solving a Continuous BAMDP. Existing discrete solvers are inapplicable, necessitating the development of Continuous BAMCP to solve these problems effectively.
>
> **Necessity of Policy Learning** Continuous BAMCP alone only yields planning results for specific states, which is insufficient for real-time control systems that require immediate inference. To enable practical deployment, we must distill the planning outcomes into a parametric policy. We clarify that both BA-MCTS and BA-MCTS-SL are policy learning methods (differing only in their update mechanisms). Therefore, policy learning is not optional; it is the essential bridge between our sophisticated planning algorithm and real-time execution.

---

### Meta-Review · Area_Chair_ENEa · 2026-01-09

**Summary:**

The decision to accept this paper is informed by the reviewers' recognition of the novelty and empirical strength of the proposed framework, which formulates offline model-based reinforcement learning as a Bayes-Adaptive MDP solved via Continuous Monte-Carlo Tree Search. The primary concerns initially raised by the reviewers focused on the conceptual validity of performing Bayesian updates on an ensemble of learned models without new online data, the clarity of the mathematical notation regarding augmented states and beliefs, and the computational cost associated with the search planning. Additionally, there was skepticism from one reviewer regarding the necessity of the search component compared to standard pessimistic Dyna-style baselines and whether the dense presentation obscured the method's practical utility.

**Reviewer Concerns:**

The rebuttal phase was highly effective in addressing the substantive technical and conceptual concerns raised by the committee. The authors successfully clarified the "offline Bayesian update" paradox for Reviewer jrky by distinguishing between belief calibration on real history and maintaining temporal consistency during imaginary rollouts, and they resolved Reviewer JBcC's confusion regarding notation and experimental discrepancies, leading to that reviewer explicitly stating they had no further significant concerns. The authors also adequately addressed Reviewer Gn3V's questions on scalability by clarifying that the computationally expensive search is restricted to offline training, leaving the deployment policy lightweight. The only concern that remains technically "outstanding" is the subjective improvement of the paper's exposition and the inclusion of the promised schematic diagrams, as Reviewer u5So did not participate in the discussion to verify if the proposed changes fully resolved their readability issues, though the authors provided strong objective evidence (ablation studies) to counter that reviewer's skepticism about the method's utility.

**Reviewer Scores:**

If all reviewers had been able to fully participate in the discussion following the rebuttal, it is highly probable that the scores would have trended upward to a clear acceptance. Reviewer JBcC, who initially gave a 6, explicitly confirmed that their main issues were addressed. Reviewer jrky, also sitting at a 6, would likely have increased their score to a 8, given that the authors provided a rigorous and logical answer to their primary conceptual question regarding the validity of the belief updates. Finally, Reviewer u5So, who assigned a 4, would likely have raised their score to at least a 6 upon realizing that their assumption about the method being no better than standard Dyna baselines was factually incorrect based on the ablation results in Table 11, and that the inference-time latency concerns were based on a misunderstanding of the system architecture.

---

### Decision · Program_Chairs · 2026-01-26

Accept (Poster)